# Using modified DNDC biogeochemical model to optimize field management of multi-crop (cotton, wheat, maize) system: a site-scale case study in northern China

Wei Zhang [1], Chunyan Liu [1], Xunhua Zheng [1,2], Kai Wang [1], Feng Cui [1,a], Rui Wang [1], Siqi Li [1,2], Zhisheng Yao [1], Jiang Zhu [1]

[1] State Key Laboratory of Atmospheric Boundary Layer Physics and Atmospheric Chemistry, Institute of Atmospheric Physics, Chinese Academy of Sciences, Beijing 100029, P. R. China

[2] College of Earth and Planetary Science, University of Chinese Academy of Sciences, Beijing 100049, P.R. China

[a] Now at the Environmental monitoring station of Jiulongpo district, Chongqing, 630050, P. R. China

*Corresponding to*: Xunhua Zheng (xunhua.zheng@post.iap.ac.cn)

**Abstract** It is still a severe challenge to optimize the field management practices for multi-crop system simultaneously aiming at yield sustainability and the minimum negative impacts on climate and qualities of atmosphere and water. This site-scale case study devoted to develop a biogeochemical process model-based approach as a solution to this challenge. The best management practices (BMPs) of a three-crop system growing cotton and winter wheat-summer maize (W-M) in rotation, which is widely adopted in northern China, were identified. The BMPs were referred to the management alternatives with the lowest negative impact potentials (NIPs) among the scenarios satisfying all given constraints. The independent variables to determine the NIPs and those as constrained criteria were simulated by the DeNitrification-DeComposition model modified in this study. Due to the unsatisfactory performance of the model in daily simulations of nitric oxide (NO) emission and net ecosystem exchange of carbon dioxide (NEE), the model was modified to include (i) newly parameterizing the soil moisture effects on NO production during nitrification, and (ii) replacing the original NEE calculation approach with an algorithm based on gross primary production. Validation of the modified model showed statistically meaningful agreements between the simulations and observations in the cotton and W-M fields. Three BMP alternatives with overlapping uncertainties of simulated NIPs were screened from 6000 management scenarios randomly generated by the Latin hypercube sampling. All these BMP alternatives adopted the baseline (currently applied) practices of rotation pattern (3 consecutive years of cotton rotating with 3 years of W-M in each 6-year cycle),

fraction for crop residue incorporation (100%), and deep tillage (30 cm) for cotton. At the same time,
these BMP alternatives would use 18% less fertilizer nitrogen and sprinkle or flood-irrigate ~23% less
water than the baseline while adopting reduced tillage (5 cm) for W-M. Compared to the baseline
practices, these BMP alternatives could simultaneously sustain crop yields, annually enlarge soil
organic carbon stock by 4‰ or more, mitigate the aggregate emission of greenhouse gases, NO release,
ammonia volatilization, and nitrate leaching by ~7%, ~25%, ~2% and ~43%, respectively, despite ~5%
increase in $N_2O$ emission. However, further study is still necessary for field confirmation of these BMP
alternatives. Nevertheless, this case study proposed a practical approach to optimize multi-crop system
management for simultaneously achieving multiple United Nations Sustainable Development Goals.
**1 Introduction**
Globally, fiber crops (i.e., cotton) and cereals such as wheat and maize have long played a relevant
role in human society because they are primary sources of materials for the textile and food industries.
In China, although cotton cultivation only covers 2.0−3.9% of the annual crop harvest area (there was a
cotton lint production of 5.3−7.6 million metric tons from 2007−2016), the cultivation of cereals is
quite large. Wheat and maize accounted for 39% and 26% of the harvest area and represented 129 and
220 million metric tons of grain, respectively, in 2016 (China Statistical Yearbook, 2017).
Northern China is both the second most important area of cotton production and the largest region
of the winter wheat-summer maize double-cropping system (i.e., both crops are harvested within a year,
and they are hereinafter referred to as W-M) in the country (e.g., Cui et al., 2014). Crop rotations of
cotton and W-M have commonly been grown in this region, alternating every 3−5 years (e.g., Liu et al.,
2010, 2014). During the last few decades, the yields of cotton, wheat and maize have been increased by
employing intensified agricultural management practices, such as increased fertilizer inputs, advanced
irrigation methods and so on (e.g., Han, 2010). A recent study indicated that the cotton cropping system
in northern China persistently functioned as an intensive carbon or net aggregate greenhouse gas (GHG)
source compared to W-M (Liu et al., 2019). These previous studies have revealed that the change in the
storage of soil organic carbon (ΔSOC), net ecosystem aggregate GHG emissions (NEGE) and other
biogeochemical processes involving the multiple cropping system in northern China are likely closely
related to the rotation pattern of cotton and W-M (e.g., Liu et al., 2010, 2014, 2019; Lv et al., 2014).
To maintain high productivity, the three-crop rotation system for cotton and W-M in northern
China are characterized by large additions of synthetic nitrogen fertilizers and irrigation water (e.g.,
Chen et al., 2014; Galloway et al., 2004), at 60−140 and 550−600 kg N ha$^{-1}$ yr$^{-1}$, and 140−200 and
90−690 mm yr$^{-1}$ for cotton and W-M, respectively (e.g., Ju et al., 2009; Liu et al., 2014; Wang et al.,
2008). High nitrogen and water inputs can result in high release potentials for nitrogenous pollutants,
and they can induce a series of environmental problems, such as increased nitrate ($NO_3^-$) leaching for
water pollution (e.g., Collins et al., 2016). In addition, other field management practices, e.g., tillage
and crop residue treatment, can also affect the emissions of reactive nitrogen and contribute to negative
environmental effects (e.g., Zhang et al., 2017; Zhao et al., 2016). Therefore, the evaluation of the
multiple-cropping system (e.g., rotations of cotton and W-M) is shifting from a single-goal method
aimed at increasing crop yields to a multi-goal approach (e.g., Cui et al., 2014; Garnett et al., 2013;
Zhang et al., 2018). A multi-goal strategy aims to simultaneously sustain/increase crop productivity to
ensure food security, increase SOC content to improve soil fertility, mitigate NEGE to alleviate climate
warming, reduce ammonia ($NH_3$) volatilization and nitric oxide (NO) emission to secure air quality,
and abate $NO_3^-$ leaching to protect water quality.
According to the multi-goal approach, an objective method is applied to identify the best
management practice (BMP), which evaluates each decision variable with price-based proxies or other
measures and screens the best option with the minimal negative impact potential (NIP) under the given
constraints at the annual scale (e.g., Cui et al., 2014; Xu et al., 2017). To screen the BMP, it is essential
to quantify the biogeochemical effects of management practices at the annual scale. Field experiments
are often capable of focusing on only the decision variables of very few management practices during
short periods (e.g., Ding et al., 2007; Liu et al., 2010, 2015; Wang et al., 2013a, b). However, this
limitation of field experiments can be overcome potentially by process-oriented biogeochemical
models, such as DeNitrification-DeComposition (DNDC) (Li et al., 1992; Li, 2000, 2007, 2016),
DAYCENT (Delgrosso et al., 2005), and LandscapeDNDC (Haas et al., 2012).
A three-crop (cotton, winter wheat and summer maize) system in southern Shanxi Province was
selected for this model-based site scale case study. This study was to (i) diagnose problems of
DNDC95 model version that has been validated in Cui et al. (2014) against the comprehensive field
measurements of the selected W-M fields, (ii) make modifications and then validate the modified
model for both cotton and W-M cropping systems, especially for the variables to determine NIPs and
those involved in given constraints; and (iii) investigate the biogeochemical effects of various rotation
patterns with different field management practices, and then, identify the multi-goal BMP alternatives
based on the modified model simulations. These efforts were undertaken to test two hypotheses. One is
that a validated process-oriented biogeochemical model is capable of addressing a challenging issue –
optimization the field management practices of a three-crop rotation system. The other hypothesizes
that the field managements of an intensive three-crop rotation system can be optimized to
simultaneously sustain the current crop yields, annually increase 4‰ of the SOC stock so as to
implement the International "4 Per 1000" Initiative (https://www.4p1000.org/) – an action to the Paris
Agreement on combating climate change, mitigate aggregate greenhouse gas emission and reduce other
negative impacts on the environment.
**2 Materials and methods**
**2.1 Brief introduction to DNDC95**

The original DNDC95 model used by Cui et al. (2014) and in this study is one of the latest DNDC

versions (www.dndc.sr.unh.edu/model/GuideDNDC95.pdf). This model consists of two components
with six modules in total. Driven by the given primary ecological factors, the former component
simulates the field states of a soil-plant system, such as the soil chemical and physical status,
vegetation growth and organic matter decomposition. Driven by the soil-regulating variables yielded by
the former component, the latter component simulates the core biogeochemical processes of carbon and
nitrogen transformations and the physical processes of liquid and gas transportations and thus the
annual dynamics of net ecosystem exchanges of carbon dioxide ($CO_2$) (NEE); emissions of methane
($CH_4$), nitrous oxide ($N_2O$), $NH_3$ and NO; and $NO_3^-$ leaching and the inter-annual dynamics of SOC
and NEGE. These features enable the model to investigate the integrative biogeochemical effects of the
rotation patterns of multiple crops and/or other management practices based on comprehensive
validation. The minimum inputs used to facilitate the model simulation include (i) the meteorological
variables of daily precipitation and maximum/minimum temperature; (ii) the soil (cultivated horizon)
properties of the clay fraction, bulk density, SOC content and pH; (iii) the crop parameters for the yield
potential, thermal degree days (TDD) for maturity, and the mass fractions and carbon-to-nitrogen (C/N)
ratios of the grain, root and leaf plus stem; (iv) the management practice variables of sowing and
harvest (dates), fraction of incorporated/retained residue at harvest, tillage (date and depth), irrigation
(date, method and water amount), and fertilization (date, type, method, nitrogen amount and C/N ratio
of organic manure); and (v) other variables (annual means of the $NH_3$ concentration in the atmosphere
and the ammonium plus $NO_3^-$ concentration in rain water). For more details about the model, please
see Li et al. (1992) and Li (2000, 2007, 2016).
**2.2 Problem diagnosis and model modification**
The daily simulations of the original DNDC95 showed poor model performance for NO emissions
from the cotton field, e.g., with a very negative Nash-Sutcliffe efficiency index (NSI) of –1.03 for the
333 observations in 2009. Meanwhile, the original model often failed to capture the daily high NEE
fluxes on rainy or cloudy days despite the agreement between the simulation and observation of the
annual cumulative NEE (Cui et al., 2014). According to the program codes, a constant fraction (0.003)
of a nitrification rate ($F_n$, in g N m$^{-2}$ d$^{-1}$) was adopted in the original model to calculate the daily NO
production in the nitrification process. This was found to account for the former problem as the fraction
could vary with soil moisture, mechanically similar with the $N_2O$ production in nitrification (shown by
the model program codes). The later problem was ascribed to the adopted algorithm to calculate daily
NEE. In the original model, a daily NEE flux was calculated as the residue of daily $CO_2$ release by soil
heterotrophic respiration and daily $CO_2$ uptake by the increase in cumulative net primary production
(NPP). The daily cumulative NPP was calculated by portioning the input of potential crop yields to the
day following a given NPP growth curve (shown by the model program codes; Li, 2016). Consequently,
the insensitivity of a daily NPP increase to radiation intensity reduction resulted in a more negative
daily NEE on a rainy or cloudy day. The model was modified in this study, as follows, to solve these
two problems.
In the model version modified in this study, the effect of the soil moisture (SM) in water-filled
pore space (WFPS, dimensionless 0−1 fraction) on NO production was parameterized by referring to
that for $N_2O$ production during nitrification and incorporated into the function by replacing the constant
fraction mentioned above (Eq. 1). This modification was adopted to reflect that high soil moisture
facilitates the production of NO as a by-product in nitrification ($NO_p$, in g N m$^{-2}$ d$^{-1}$). The maximum
fraction of NO production ($K_n$, dimensionless 0−1 fraction) during nitrification was calibrated as 0.03
using the observed daily NO fluxes from October 2007 to October 2008. The other observations of
daily and annual NO fluxes from both adjacent lands with different field treatments (Table S1) were
used to validate this modification.

$$NO_p = \quad SM^{5.0} K_n F_n \tag{1}$$

In the model version modified in this study, a daily NEE flux (g C m$^{-2}$ d$^{-1}$) is calculated as the
residue of the daily $CO_2$ release from ecosystem respiration (ER, in g C m$^{-2}$ d$^{-1}$) and daily $CO_2$ uptake
due to gross primary production (GPP, in g C m$^{-2}$ d$^{-1}$). While the daily ER is simulated as it is in the
original model, the daily GPP is calculated using Eq. 2 that is a widely applied hyperbola function of
photosynthetically active radiation (PAR, in mmol m$^{-2}$ d$^{-1}$) (e.g., Wang et al., 2013a; Zheng et al.,

2008).

$$GPP = \alpha PAR\ GPP_{max}/\ (\alpha PAR + GPP_{max}) \tag{2}$$

The photosynthetic parameters in Eq. 2, $\alpha$ (g C mmol$^{-1}$) and $GPP_{max}$ (g C m$^{-2}$ d$^{-1}$), denote
apparent quantum yield and maximum GPP, respectively. Each parameter is quantified as the product
of shoot standing biomass ($B_s$, in g C m$^{-2}$) and biomass-specific apparent quantum yield ($f_1$, in g C
mmol$^{-1}$ (g C m$^{-2}$)$^{-1}$) or specific $GPP_{max}$ ($f_2$, in g C m$^{-2}$ d$^{-1}$ (g C m$^{-2}$)$^{-1}$) corrected by an adaptation factor
($a$, dimensionless) reflecting inter-annual variations of a crop (Eqs. 3−4). The variable $B_s$ is simulated
as it is in the original model. The seasonal dynamics of $f_1$ and $f_2$ is a function of normalized plant
developing stage (ds, dimensionless 0−1 fraction) (Eqs. 5−6). The functions of $f_1$ and $f_2$ take the forms
presented by Zheng et al. (2008) for winter wheat while their empirical parameters of $a$, $b$, $c$, $d_1$, $d_2$, $g$, $h$,
$i$, $j$, $l$ and $m$, can be calibrated to adapt both functions to given conditions. Two daily NEE fluxes per
week were randomly selected from the year-round eddy covariance observations in both cotton and
W-M cropping systems (Cui et al., 2014; Liu et al., 2019; Wang et al., 2013a) to calibrate the values of
these parameters specifically for cotton, winter wheat and summer maize, while remaining daily data
(independent NEE observations) were used to evaluate this modified NEE algorithm. For the calibrated
values of these parameters, refer to Table S2.

$$\alpha = af_1B_s \tag{3}$$

$$GPP_{max} = af_2B_s \tag{4}$$

$$f_1(ds) = be^{-cds}\ (ds \ge d_1) \tag{5}$$

$$f_1(\mathrm{ds}) = ge^{h\mathrm{ds}} \ (\mathrm{ds} < d_1)$$

$$f_2(\mathrm{ds}) = ie^{-j\mathrm{ds}} \ (\mathrm{ds} \geq d_2)$$

$$f_2(\mathrm{ds}) = l\mathrm{ds} + m \ (\mathrm{ds} < d_2)$$

(6)

**2.3 Brief introduction to the selected field site and information of the data for model evaluation**

The field site (34°55.50′N, 110°42.59′E and altitude of 348 m) selected for this modeling case study is located at Dongcun Farm, near Yongji County, Shanxi Province, in northern China. The site is subject to a temperate continental monsoon climate, and it had an annual precipitation of 580 mm and a mean air temperature of 14.4 °C from 1986−2010 (Cui et al., 2014). Cotton, winter wheat and summer maize are the major crops grown at this farm and the surrounding regions. Field experiments were performed on two adjacent lands (each of which was 100 m wide and 200 long) for cotton and W-M in 2007−2010. The soil of the land cultivated with cotton and W-M was clay loam, with approximately 38% and 32% clay (< 0.002 mm), 57% and 50% silt (0.002−0.05 mm), 5% and 18% sand (0.05−2 mm), 10.0 and 11.3 g kg$^{-1}$ SOC, 1.1 and 1.1 g kg$^{-1}$ total nitrogen and pH (in H$_2$O) of 8.0 and 8.7 at a 0−10 cm depth and bulk densities (0−6 cm depth) of 1.20 and 1.17 g cm$^{-3}$, respectively (Liu et al., 2010, 2011, 2012). A sprinkler system was applied to both lands. For more detailed information on the field experiments and observed data, please refer to Cui et al. (2014), Liu et al. (2010, 2011, 2014, 2015) and Wang et al. (2013a, b).

The modified model was validated with observations in both lands. The daily meteorological data from 2004−2010 were obtained directly from Cui et al. (2014). The measured data were used directly for the minimum required soil properties. The input data on the field capacity and wilting point in WFPS were 0.65 and 0.2, respectively (Cui et al., 2014). The crop parameters for cotton were directly determined by the field measurements, which were 1900 kg C ha$^{-1}$ for potential grain yield (1.2 times the mean of the measured values), 0.41 and 25, 0.16 and 40, and 0.43 and 40 for the mass fractions and C/N ratios of the grain, root and leaf plus stem, respectively, and 3600 °C for the TDD. Detailed management practices (Table S3) were obtained from Li et al. (2009) and Liu et al. (2014). Different from the locally conventional fertilizer application rate of 110−140 kg N ha$^{-1}$ yr$^{-1}$ for cotton, the fertilizer doses for the experimental cotton field in 2007 and 2008 were reduced to 66−75 kg N ha$^{-1}$ yr$^{-1}$ to avoid the overgrowth of the leaves in place of seeds or lint. The data of the cotton field available for the model calibration and validation included the daily observed soil (5 cm depth) temperature,

topsoil (0−6 cm) moisture, daily NEE from eddy covariance measurement from 2008 to 2009,
sub-weekly observed $CH_4$ fluxes in the growing season of 2010 (Liu et al., 2010, 2014, 2019; Wang et
al., 2013b), daily $N_2O$ and NO fluxes from 2007 to 2008, annually measured grain yields from 2008 to
2010, and annually $NO_3^-$ leaching from 2008 to 2009. All the data used by Cui et al. (2014) in
validation of the original model were used to re-validate the modified model performances for the
selected W-M fields with different field managements. In addition, two observations of the cumulative
$NH_3$ volatilizations following urea topdressing to the winter wheat in the spring of 2008 (Tong et al.,
2009) were also involved in evaluation of the model performance (Yang et al., 2011). The information
for all the data used in the modified model calibration and validation are detailed in Table S1.
**2.4 Scenario settings and simulations**

For the investigated three-crop system, this study attempted to identify the BMP alternatives with

the best rotation pattern between cotton and W-M and the optimized field management practices of this
rotation pattern. For this purpose, six rotation pattern scenarios (hereinafter referred to as $R_0$, $R_1$, ..., $R_5$)
were set for a 6-year cycle that was widely applied by the local farmers (Liu et al., 2010, 2011, 2014).
The subscript number of each rotation pattern represents the number of the consecutive years with
cotton cultivation. For instance, $R_0$ denotes a 6-year monoculture of W-M, and $R_2$ the rotation pattern
with 2-year continuous cotton rotated with 4-year of continuous W-M. Because the local farmers
typically did not adopt cotton monoculture for longer than five years, the longest cotton monoculture
lasted for only 5 years ($R_5$). The transitions between cotton and W-M in the scenario rotations are
detailed in Table S4.

As for the setting of the field management scenarios for the individual rotation patterns, four field

management factors were considered, which were (i) nitrogen fertilizer dose, (ii) water amount (iii)
method of irrigation, and (iv) depth of tillage. The values of these factors used for the baseline scenario
were known as the observations for the conventional management practices in the experimental region
(Tables S3, S4 and S5). The nitrogen doses of the baseline were 110 and 430 kg N $ha^{-1}$ $yr^{-1}$ for the
cotton and W-M, respectively. Over the last few decades, the fields in this region have been mostly
flood-irrigated (Liu et al., 2010). Thus, flood-irrigation was chosen as the baseline method. The
baseline timings and water amounts were established by referring to the 10- to 30-d cumulative
precipitation prior to the individual irrigation events and the recorded timings and water amounts of the
conventional management practices in the two adjacent lands. Thus, the irrigation frequencies and
annual cumulative water amounts of the baseline varied from 1 to 3 times and 75 to 230 mm yr$^{-1}$ for
cotton and 4 to 6 times and 290 to 510 mm yr$^{-1}$ for W-M (Table S5). The fraction of 100% for residue
incorporation and the conventional tillage to a depth of 20 for W-M and 30 cm for cotton were applied
for the baseline practices. To screen the BMPs by fully taking into account the independent and
interactive effects of rotation patterns and field management on the NIPs, totally 6000 field
management scenarios (each being a combination of the four management factors) for all the six
rotation pattern scenarios (1000 for each) were randomly generated. The fertilizer doses and irrigation
water amounts were randomly selected within their lower and upper bounds of continuous variations,
using the Latin hypercube sampling. The lower and upper bounds for nitrogen fertilizer doses (44−172
and 110−430 kg N ha$^{-1}$ yr$^{-1}$) and irrigation water amount (40−100 mm per event) were set as 40% and
100% of the baseline, respectively. Only two irrigation methods, flooding (IF) as the baseline and
sprinkling (IS), were included for the random sampling of this management factor. For cotton, the
tillage depth was fixed at 30 cm. For W-M, four tillage depths (0 cm for no-till, 5 and 10 cm for
reduced tillage, and 20 cm for the conventional practice) were included for random sampling of this
factor. The BMPs for each rotation pattern scenario were first screened from 1000 field management
scenarios. Then, the BMP alternatives were finally screened from these BMPs of the individual rotation
pattern scenarios. These identified BMP alternatives would include the scenarios with overlapping
uncertainties of NIPs, for which the random error at one times standard deviation (SD, instead of two
times SD) for the total simulation error was adopted for the NIP of each scenario so as to achieve a
relatively high screening precision. The decision variables and NIP of baseline management scenarios
for each rotation pattern scenario were used to particularly address the biogeochemical effects of
rotation pattern.
An 18-year simulation was performed for each scenario. The annual averages for the simulated
yields, decision variables and NIPs were used to address the biogeochemical effects of rotation patterns
or to screen the BMP alternatives. The simulations of all scenarios were driven by the meteorological
data observed at the Yuncheng station (approximately 60 km east to the experimental site) from
1996−2013 (http://data.cma.cn/data/cdcindex/cid/6d1b5efbdcbf9a58.html). To stabilize the carbon and
nitrogen dynamics and reduce the residual effects of the initial conditions (Palosuo et al., 2012; Zhang
et al., 2015), a 12-year spin-up for each scenario was performed (i.e., a period of two consecutive
6-year rotation cycles) before the 18-year simulation. The spin-up for each scenario was driven by the
same rotation pattern and field management practices as this scenario.
**2.5 Method for identifying the best management practices**
An objective method jointly relying on three constraints and NIPs was adopted in this study to
screen the BMPs from given scenarios. These constraints included (i) stable or increased crop yields, (ii)
annually increased SOC stock by 4‰ or more, and (iii) reduced NEGE by 5% or more. In the present
study, the NEGE was determined by summing up the emissions of $CH_4$ and $N_2O$ and $-\Delta SOC$, which
was quantified as a $CO_2$ equivalent ($CO_2eq$) quantity. In the quantification of a NEGE, the global
warming potentials at 100-year time horizon, 34 for $CH_4$ and 298 for $N_2O$ (IPCC, 2013), were used to
convert the quantities of both gases into $CO_2$ equivalents. A NIP was expressed as a price-based proxy
quantity in US$ $ha^{-1}$ $yr^{-1}$ and used to evaluate the potential for a climatically and environmentally
integrative impact exerted by a set of field management practices for multi-crop system. The NIP was
determined as a linear function of the individual decision variables, following Eq. 7, wherein the
multi-goal decision variables, NEGE, $NH_3$, NO, $N_2O_{ODM}$, and NL, represent the annual net ecosystem
aggregate GHG emission (Mg $CO_2eq$ $ha^{-1}$ $yr^{-1}$), $NH_3$ volatilization, NO emission, release of $N_2O$ as an
ozone layer depletion matter, and hydrological nitrogen loss (mainly by $NO_3^-$ leaching), respectively
(kg N $ha^{-1}$ $yr^{-1}$ for all the nitrogen-based variables). The coefficients $k_1$, $k_2$, $k_3$, $k_4$ and $k_5$ are mass-scaled
price-based proxies for the NEGE, $NH_3$, NO, $N_2O_{ODM}$, and NL, with values of 7.00 US$ $Mg^{-1}$ $CO_2eq$
and 5.02, 25.78, 1.33 and 1.92 US$ $kg^{-1}$ N, respectively (Cui et al., 2014). A lower NIP indicated a
better set of management practices that can exert smaller negative impacts on the climate and
environment. Accordingly, the BMP was identified as the scenario with the lowest NIP among the
scenarios that could satisfy all three constraints.

$$NIP = k_1 NEGE + k_2 NH_3 + k_3 NO + k_4 N_2O_{ODM} + k_5 NL \qquad (7)$$

**2.6 Method for uncertainty quantification**
The model simulation error ($\varepsilon_s$) of a NIP, a constraint variable (e.g., crop yield) or a decision
variable involved in Eq. 7 represented the total simulation uncertainty. It was made of two components.
One was the input-induced uncertainty ($\varepsilon_{input}$) due to the uncertainties of input items; and the other was
the model uncertainty ($\varepsilon_{\text{model}}$) due to insufficiencies in scientific structure or process parameters.
The $\varepsilon_{\text{input}}$ of a simulated variable was a random error if the uncertainties of model input items were
known as random errors. It was estimated using the Monte Carlo test with the Latin hypercube
sampling within the uncertain ranges (95% confidence interval (CI)) of sensitive input items. In DNDC,
the four basic soil properties (bulk density, pH, clay fraction and SOC content) as input items were
sensitive to the model outputs (e.g., Li, 2016). Accordingly, the uncertainties of these soil properties
were regarded to be the major dominators for the uncertainties of the model outputs, such as the
constraint/decision variables, or NIP. The initial bulk density, pH, clay fraction and SOC content ranged
$1.13-1.25$ g cm$^{-3}$, $8-8.7$, $0.31-0.39$ and $9-12$ g kg$^{-1}$ (at the 95% CI), respectively, which were adapted
from the means and two times SD of spatially replicated observations in the two adjacent lands (Liu et
al., 2014, 2019). A uniform distribution for each of these soil properties was assumed in Monte Carlo
test, in which the sampling and simulation were iterated until the mean of simulated NIPs for all
iterations converged to a certain level within a tolerance of 1%. The NIP uncertainty due to the model
input uncertainties was presented as the SD of these iterated simulations.
An $\varepsilon_{\text{s}}$ was systematic error reflecting a model simulation bias diverging from the truth. In this
regard, the $\varepsilon_{\text{s}}$ of a constraint/decision variable could be estimated using the slope of a zero-intercept
univariate linear regression of simulations against observations (ZIR$_{\text{s-o}}$) or model relative biases (MRBs)
resulted from model validation (Eq. 8). The MRBs were used only in case a significant ZIR$_{\text{s-o}}$ was
failed to be obtained in model validation. To ensure a relatively high BMP screening precision, the
random uncertainty of the $\varepsilon_{\text{s}}$ of a NIP was presented as one times SD, instead of the two times SD to
represent 95% CI, as the BMP alternatives were referred to the management scenarios with overlapping
NIP uncertainties. For a constraint/decision variable, the mean or SD of $\varepsilon_{\text{s}}$ in an absolute magnitude
was estimated as the product of (i) an adjusting factor, (ii) the simulated variable quantity, and (iii) an
error factor. The adjusting factor was obtained from model validation, which was estimated as the mean
of the ratios of individual observations to simulations. The error factor for a variable with a significant
ZIR$_{\text{s-o}}$ was given as (Mean-Slope$_{\text{s-o}}-1$) $\pm$ SD-slope$_{\text{s-o}}$, wherein Mean-Slope$_{\text{s-o}}$ and SD-slope$_{\text{s-o}}$ denote the
mean and SD of the ZIR$_{\text{s-o}}$ slope, respectively. The item prior to and following the "$\pm$" was used to
estimate the mean and SD of the $\varepsilon_{\text{s}}$. For a variable failed to obtain a significant ZIR$_{\text{s-o}}$ in model
validation, the mean and SD of the error factors were given as the mean and SD of the MRBs. The
mean of the $\varepsilon_s$ for a NIP was estimated by simply summing up the weighted absolute $\varepsilon_s$ of individual
decision variables. This was because the decision variables involved in the additive items of Eq. 7 were
independent from each other. Meanwhile, the SD of the $\varepsilon_s$ for a NIP was mathematically propagated
from the SDs of the absolute $\varepsilon_s$ of the decision variables.
In this study, the constraint variables included crop yield, $-\triangle$SOC and NEGE while NEGE was
also one of the decision variables. Although there was no direct measurement of $-\triangle$SOC and NEGE,
their observation-oriented estimates were involved in the model validation, which provided the basis
for the $\varepsilon_s$ estimation of either variable. For the experimental fields with NEE observations, there was no
significant input quantity of organic matter in manure or any other form while crop residues were fully
incorporated into the soil. In these cases, each annual/seasonal $-\triangle$SOC could be estimated as the sum
of annual/seasonal NEE and yields, according to the mass conservation law, and used to represent the
observation, with its uncertainty propagated from the random errors of the annual/seasonal yield and
NEE measurements. The random error of this observation-oriented $-\triangle$SOC that represented the
annual/seasonal net $CO_2$ emission from the cropping system and those of the observed annual
cumulative $CH_4$ and $N_2O$ were propagated to estimate the observational error of the annual/seasonal
NEGE. These observation-oriented estimates of $-\triangle$SOC or NEGE were involved in model validation.
**2.7 Statistics and analysis**
Statistical criteria simultaneously used to evaluate the model validity included (i) the index of
agreement (IA) (Eq. 9), (ii) the NSI (Eq. 10) (e.g., Moriasi et al., 2007; Nash and Sutcliffe, 1970), (iii)
the determination coefficient ($R^2$) (Eq. 11) and slope of a ZIR of observations against simulations
(Jiang, 2010; Li et al., 2019), and (iv) the MRB (Eq. 8) (e.g., Congreves et al., 2016; Willmott and
Matsuura, 2005). In Eqs. (8−11), $k$ and $n$ ($k = 1, 2..., n$) denote the $k$th pair and the total pair number of
the values, respectively, and $\bar{o}$ represents the mean of the observations ($o$), respectively, and $\hat{o}$ is the
predictions using the ZIR. The IA index fell between 0 and 1, with a value closer to 1 indicating a
better simulation, and vice versa. An NSI value between 0 and 1 indicated acceptable model
performance. Better model performance was indicated by a slope and an $R^2$ value those were closer to
1, and vice versa. An |MRB| value smaller than the double coefficients of variation (CVs) of replicated
observations implicated a valid simulation (Dubache et al., 2019).

$$\text{MRB} = \frac{s_k}{o_k} - 1 \tag{8}$$

$$\text{IA} = 1 - \frac{\sum_{k=1}^{n}(s_k - o_k)^2}{\sum_{k=1}^{n}(|s_k - \bar{o}| + |o_k - \bar{o}|)^2} \tag{9}$$

$$\text{NSI} = 1 - \frac{\sum_{k=1}^{n}(o_k - s_k)^2}{\sum_{k=1}^{n}(o_k - \bar{o})^2} \tag{10}$$

$$R^2 = 1 - \frac{\sum_{k=1}^{n}(o_k - \hat{o}_k)^2}{\sum_{k=1}^{n}(o_k - \bar{o})^2} \tag{11}$$

In this study, the statistical analysis and graphical comparison were performed with the SPSS
Statistics Client 19.0 (SPSS Inc., Chicago, USA) and Origin 8.0 (OriginLab, Northampton, MA, USA)
software packages.
**3 Results**
**3.1 Validation of daily simulations for both cropping systems**
The seasonal dynamics and magnitudes of the soil (5 cm) temperature and topsoil (0−6 cm)
moisture were predicted well by the model simulations (Figs. 1a−b). The sound model performance
was indicated by the IA, NSI, and ZIR slope and $R^2$ values of 0.98 and 0.83, 0.93 and 0.15, 0.93 and
0.83, and 0.95 ($n = 677$, $p < 0.001$) and 0.42 ($n = 432$, $p < 0.001$) for the temperature and moisture,
respectively.
As compared to the original model, the modified model showed much better performances in
simulating daily NEE fluxes from both cropping systems. It particularly well simulated the abrupt NEE
fluxes on cloudy or rainy days in the growing season (Figs. 1c−e). In comparison with the original
model for daily NEE simulations, the modified model enhanced the IA, NSI, and ZIR slope and $R^2$
from 0.74 to 0.81, 0.32 to 0.60, 0.60 to 0.96 and 0.27 to 0.60 ($n = 261$, $p < 0.001$), respectively, for the
cotton field, and from 0.75 to 0.80, 0.30 to 0.51, 0.69 to 0.80 and 0.45 to 0.55 ($n = 311$, $p < 0.001$),
respectively, for the W-M field. For the $CH_4$ uptake, the observations and simulations showed similar
seasonal variations (Fig. 1f), with the IA, NSI and ZIR slope and $R^2$ of 0.68, < 0, 0.70 and 0.08 ($n = 69$,
$p = 0.018$), respectively.
The simulated seasonal patterns and peak emissions of $N_2O$ and NO generally matched the
observations (exemplified by Figs. 1g−h for the cotton field). In comparison with the original model for
daily $N_2O$ flux simulations, the modified model performed comparably well for the cotton field, with
NSI of around −0.45, ZIR slope ~0.49 and $R^2$ of ~0.39 ($n = 592$, $p < 0.001$), while it showed better
performance for the W-M fields, with IA, NSI and ZIR slope and $R^2$ of 0.69 versus 0.52, 0.03 versus
−0.26, 0.62 versus 0.46 and 0.16 ($n = 976$, $p < 0.001$) versus "not available", respectively. Relative to
original model, the modified model showed improved simulations for the daily NO fluxes from the
cotton field, with increased IA, NSI, and ZIR slope and $R^2$ from 0.62 to 0.78, −1.03 to −0.04, 0.37 to
0.54 and 0.09 to 0.39 ($n = 333$, $p < 0.001$), respectively, while it performed comparably well for those
from the W-M fields, with IA, NSI, and ZIR slope and $R^2$ of ~0.82, ~0.32, ~0.74 and 0.35−0.40 ($n =$
967, $p < 0.001$), respectively.
In comparison with the observed daily $NH_3$ fluxes (measured using micrometeorological
technique) following single fertilization event of the maize season in 2008, the modified model
simulations showed IA, NSI, and ZIR slope and $R^2$ of 0.87, 0.12, 0.68 and 0.53 ($n = 11$, $p = 0.07$),
respectively. However, the model failed to capture the immediate responses of daily $NH_3$ fluxes to the
urea addition to the wheat fields, which were measured using a quasi-dynamic chamber method.
**3.2 Validation of simulated variables in annual/seasonal cumulative quantities**
For the cotton yields (seeds plus lint) over the three consecutive experimental years (2008−2010),
the simulations were consistent with the observations in terms of the smaller |MRBs| (0.4−24%) than
the double CVs (39−56%) of the spatially replicated measurements. For all the experimental treatments
of the cotton, wheat and maize, the simulated yields of both the modified and original model highly
agreed with observations (Fig. 2a), with IA of 0.93−0.95, NSI of 0.75, and ZIR slope and $R^2$ of
0.96−1.00 and 0.75−0.78, respectively ($n = 35$, $p < 0.001$). This validation resulted in an adjusting
factors of 0.96 and smaller error factors of 3.0 $\pm 1.6\%$ for crop yields simulated by the modified model.
For the annual cumulative NEE of the cotton field during the two consecutive year-round periods
and the seasonal cumulative NEE in two wheat seasons and one maize season, the simulations of both
model versions showed comparably significant agreements with the observations (Fig. 2b), with IA of
0.99−1.00, NSI of 0.95−1.00, and ZIR slope and $R^2$ of 0.92−1.02 and 0.97−0.99, respectively ($n = 5$, $p$
$\leq 0.000−0.002$). The modified model simulations showed |MRBs| of 6−16%, which were much less
than the reported CV (25%) of the eddy covariance observations.
As compared to the annual/seasonal NEGE quantities derived from the observations of crop yields,
annual/seasonal cumulative NEE and fluxes of $CH_4$ and $N_2O$, the simulations implicated good
performance of the modified model (Fig. 2c), with IA of 0.96, NSI of 0.77, ZIR slope and $R^2$ of 0.73
and 0.91 ($n = 5$, $p = 0.013$). Although the simulations showed an average overestimation of ~25%, their
|MRBs| were only 17−72% (33% on average) of the observation-oriented CVs (27−170%, with a mean
of 91%), implicating a statistically meaningful good performance of the modified model. This
validation resulted in an adjusting factor of 0.73 and error factors of 25 $\pm$19% for the annual/seasonal
NEGE simulated by the modified model.

In comparison with the annual/seasonal $\triangle$SOC quantities estimated from the observed crop yields

and annual/seasonal cumulative NEE, the simulations by the modified model (Fig. 2d) showed IA of
0.96, NSI of 0.75, and ZIR slope and $R^2$ of 0.71 and 0.92 ($n = 5$, $p = 0.011$). The ZIR slope indicated an
average overestimation of the model by ~30%. Nevertheless, the |MRBs| of the individual simulations
were only 4−79% (30% on average) of the observation-oriented CVs (30−210%, with a mean of 97%),
still indicating a statistically meaningful consistence. This validation resulted in an adjusting factor of
0.71 and error factors of 30 $\pm$19% for the annual/seasonal $\triangle$SOC simulated by the modified model.

The model simulations of the annual cumulative $CH_4$ uptake in 2009 and 2010 showed significant

agreements (Fig. 2e), with IA of 0.98, NSI of 0.91, and ZIR slope and $R^2$ of 1.00 and 0.91 ($n = 7$, $p <$
0.001). The |MRBs| were only 5−56% (25% on average) of the double CVs (10−24%, with a mean of
17%) for the spatially replicated observations. This validation resulted in an adjusting factor of 1.00
and error factors of −0.2 $\pm$1.7% for the modified model simulations of cumulative $CH_4$ uptake.

The modified model simulations of the annual cumulative $N_2O$ emissions from all the field

experimental treatments of the cotton and W-M fields were comparable with the observations (Fig. 2f),
showing IA of 0.94, NSI of 0.72, and ZIR slope and $R^2$ of 0.90 and 0.83 ($n = 12$, $p < 0.001$). The
|MRBs| of the individual simulations were only 6−93% (36% on average) of the double CVs (23−64%,
with a mean of 47%) for the spatially replicated observations. For the annual cumulative $N_2O$
emissions simulated by the modified model, this validation resulted in an adjusting factor of 0.90 and
error factors of 8.7 $\pm$4.5%.

As compared to annual cumulative $N_2O$ emissions, slightly better consistence with observations

was obtained for the modified model simulations of the annual cumulative NO emissions from the
cotton and W-M fields under different experimental conditions (Figs. 2f−g). The NO simulation
showed IA of 0.97, NSI of 0.85, and ZIR slope and $R^2$ of 0.90 and 0.94 ($n = 11$, $p < 0.001$). The |MRBs|
of the individual simulations were only 2−52% (23% on average) of the double CVs (30−99%, with a

mean of 66%) for the spatially replicated observations. This validation provided an adjusting factor of 0.90 and error factors of 10.1 $\pm$ 3.2% for the cumulative NO emissions simulated by the modified model.

The simulations of the cumulative $NH_3$ volatilizations during the 11 days following the three urea application events, with one in the maize field in summer and two in the winter wheat fields (with flood-irrigation and sprinkling, respectively) in spring (Fig. 2h), showed IA of 0.97, NSI of 0.86, and ZIR slope and $R^2$ of 1.00 and 0.86 ($n = 3$, $p = 0.246$). The simulations resulted in smaller |MRSs| (3.8−8.8%, −0.4% on average) than the double CVs (16−18%) for the spatially replicated measurements, despite the model failure in capture the quick responses of daily $NH_3$ fluxes to the urea top-dressing events. This validation resulted in an adjusting factor of 1.00 and error factors of −0.4 $\pm$ 7.3% for the modified model simulations of cumulative $NH_3$ volatilization following nitrogen applications.

The modified model simulations of the cumulative $NO_3^-$ leaching from cotton field in two consecutive years agreed with the observations, in terms of the smaller MRBs of −32% to −27% than the two times CVs (109−115%) for the spatially replicated observations. These MRBs represented the model-underestimations by respectively 3−4 and 13−21 kg N $ha^{-1}$ $yr^{-1}$ for the annual $NO_3^-$ leaching rates in the cotton and W-M fields subject to the currently applied field management practices. This validation derived an adjusting factor of 1.42 and error factors of −29 $\pm$ 4% for the modified model simulations.

The above results suggested that the modified DNDC95 model was especially applicable at this field site for investigating the biogeochemical effects of different rotation patterns between the cotton and W-M and those exerted by different management practices, and thus was capable of BMP identification.

**3.3 Biogeochemical effects of different cotton and wheat-maize rotation patterns**

Figure 3 illustrated the dynamics of the crop yields and each decision variable resulting from the consecutive simulations over 18 years for all the rotation patterns subject to the field management practices of the baseline scenario. Figure 4 showed the relationship between the annual average of each decision variable and the number of consecutive years of cotton monoculture within the rotation patterns.

The average grain yields for the cotton, wheat and maize were not significantly different among

the various rotation patterns, with averages of 3.5, 4.8 and 6.7 kg dry matter $ha^{-1}$ for cotton, wheat and
maize, respectively (Figs. 3a−c).

For the dynamic changes in the annual SOC stocks, the values were generally positive for the

W-M but negative for the cotton, except for the first year after the transition to this fiber crop. As
indicated by Fig. 3d, the simulated SOC contents over the 18-year period increased for $R_0$, $R_1$, $R_2$ and
$R_3$ but decreased for $R_4$ and $R_5$. The annual average −ΔSOC increased significantly ($p < 0.001$) with an
increase in the consecutive years of cotton monoculture from 0 to 5 within the 6-year rotation cycle
(Fig. 4a). The rotation patterns with the baseline management showed small variations in the $CH_4$
uptake (Fig. 3e), with the uptake rates ranging from 1.6 to 2.1 kg C $ha^{-1}$ $yr^{-1}$. However, the annual
averages for the $CH_4$ uptake increased significantly ($p < 0.001$) with the increased consecutive years of
cotton monoculture (Fig. 4b). For $N_2O$, the annual emissions showed large inter-annual variations (Fig.
3f), with CVs of 26−48%. In addition, the average emissions of this gas decreased significantly from
4.6 to 2.6 kg N $ha^{-1}$ $yr^{-1}$ (Fig. 4c) after increasing consecutive years of cotton monoculture ($p < 0.001$).
As a result, the NEGE was significantly promoted ($p = 0.002$) (Figs. 3g and 4d).

Regarding the gaseous air pollutants $NH_3$ and NO, the simulated emissions ranged from 17 to 103

and 0.5 to 3.3 kg N $ha^{-1}$ $yr^{-1}$, respectively (Figs. 3h−i). Figures 4e and f showed that the average annual
emissions of both gases were significantly reduced after increasing the consecutive years of cotton
monoculture ($p ≤ 0.001$). The annual $NO_3^-$ leaching of the different rotation patterns displayed
significant inter-annual variations (Fig. 3j), with CVs of 41−69%. Thus, the annual averages for $NO_3^-$
leaching changed insignificantly in response to the consecutive years of cotton monoculture ($p < 0.056$;
Fig. 4g).

The NIP varied significantly among the various rotation patterns ($p < 0.001$), declining from 610

to 324 US$ $ha^{-1}$ $yr^{-1}$ with increased consecutive years of cotton monoculture (Fig. 4h). For the three
constraints, the crop yields showed no obvious differences among the various rotation patterns. Both $R_0$
and $R_5$ represented the typical rotation patterns in the region. The simulations for the former indicated
the greatest increase in SOC and the lowest NEGE but the highest NIP, while those for the latter
showed the greatest SOC loss and the largest NEGE but the lowest NIP (Figs. 4a, d and h). These
patterns indicated that neither typical rotation pattern is sustainable.
**3.3 Identification of best management practices**

474        Out of the 6000 field management scenarios, three under the $R_3$ were finally identified as the BMP

alternatives, which simultaneously satisfied the given constraints while yielding the comparably lowest
NIPs (332−335 US\$ ha$^{-1}$ yr$^{-1}$) within the overlapping uncertain ranges with $\varepsilon_s$ of −22 $\pm$ 16 US\$ ha$^{-1}$
yr$^{-1}$ (Table 1). These BMP alternatives for the three-crop system recommended the following
combination of field management practices including (i) the currently applied 6-year rotation cycle
with three-year cotton monoculture rotated with three-year W-M, (ii) full incorporation of crop residues
at harvest, (iii) the presently adopted crop cultivars and timing of sowing, fertilization (date, depth and
splits), irrigation (date and times) and harvest, (iv) urea alone 18% lower rates (90 and 353 kg N ha$^{-1}$
yr$^{-1}$ for the cotton and W-M, respectively) than the conventional nitrogen fertilization, (v) sprinkling or
flood-irrigation with ~23% less water (~77 mm per event) than the conventional flood-irrigation, and
(vi) the conventional plough tillage (30 cm depth) following final cotton harvest but reduced tillage
(rotary 5 cm depth) for the W-M. In comparison with the simulations driven by the baseline scenario
($R_3$ as the currently applied rotation pattern and its field management practices), the identified BMP
alternatives could sustain the baseline crop yields while simultaneously enlarging the SOC stock by $\geq$ 4‰
and mitigating the NEGE, $NH_3$ volatilization, NO emission and $NO_3^-$ leaching by ~7%, ~25%, ~2%,
and ~43%, respectively, despite a slight increase (by ~5%) in the $N_2O$ emission (Table 1).
**4 Discussion**
**4.1 Model performance**

492        The DNDC model has been widely applied in agricultural systems around the world. The version

modified in this study showed good performance in simulating the soil environmental factors (soil
temperature and moisture), crop yields, NEE, $NH_3$ volatilization, $CH_4$ uptake, emissions of $N_2O$ and
NO, and $NO_3^-$ leaching for the investigated lands cultivated with cotton and W-M under different field
management treatments. The satisfactory validations of both crop systems, especially for the constraint
and decision variables at the annual scale, suggested that the modified DNDC95 could be applied to
quantify the constraint and decision variables to determine the NIP for the cotton and W-M rotation
system under various management practices.

500        The well-simulated soil environmental factors and crop yields provided a solid basis for further

simulating the constraint and decision variables under any field management condition of the
three-crop rotation system. This is because the soil environmental factors are the key factors regulating
the biogeochemical processes and crop yields are indicators of essential processes in plant nitrogen
uptake (Chirinda et al., 2011; Kröbel et al., 2010). For the simulations of the $N_2O$ and NO emissions,
discrepancies in daily emissions generally occur in the simulations of DNDC or other current
biogeochemical models due to the interactions among soil environmental factors and complex carbon-
and nitrogen-related processes (e.g., Bell et al., 2012; Chirinda et al., 2011; Cui et al., 2014; Lehuger et
al., 2011), which may ocassionally result in the significant time lags between the observations and
simulations (e.g., Zhang et al., 2015). For the cotton in this study, the significant underestimation of
daily NO fluxes in spring was solved to some extent through modifying the model version used by Cui
et al. (2014). However, this improvement did not significantly affect the annual cumulative emissions,
which were not mainly contributed by the spring fluxes (Liu et al., 2015). In fact, occasional time lags
of one to a very few days for measured/simulated daily fluxes seldom lead to a significant modification
for the seasonal/annual cumulative emissions of a nitrogenous gas. This is attributed to the control of
the mass conservation law and the canceling effect of negative and positive daily errors. The modified
algorithm improved the simulations of daily NEE fluxes, thus providing solid basis for yielding reliable
annual/seasonal cumulative NEE quantities. According to the mass conservation law, the annual
cumulative NEE can be involved in as one of the two additive items to estimate the annual $\triangle SOC$ of an
annual crop system with retention/incorporation of full residues but without significant input/output of
organic matter other than product removal at harvest. This approach may be used as an alternative
algorithm in the modified model to simulate the annual $\triangle SOC$ of such a cropping system. In the
present case study, the annual $\triangle SOC$ simulated by the modified model using this alternative approach
were consistent with those simulated by the algorithm that quantifies the annual $\triangle SOC$ by summing up
the annual carbon pool changes in the humus, microbial biomass and dissolvable organic compounds
(Cui et al., 2014). The consistence was indicated by the |MRBs| of 20 ± 50% versus 37 ± 117% (95%
CI) in comparison with the three observation-oriented estimates of annual $\triangle SOC$ (two for cotton and
one for the W-M). Due to the marginally small sample size ($n = 3$), this preliminary result still requires
confirmation in further study. The simulated $NH_3$ volatilizations from the cotton field accounted for
18−24% of the applied fertilizer nitrogen during the two year-round periods involved in the model
validation. These simulated nitrogen loss rates through $NH_3$ volatilization were comparable with the
reported field measurements of 10−23% (Li et al., 2016).
The model validation in this study suggested that the satisfactory simulations of constraint and
decision variables at the annual scale could provide a solid basis for BMP identification. Because of the
limited annual observations of $NH_3$ volatilization, $NO_3^-$ leaching and $\triangle$SOC estimated from annually
measured NEE, the insufficient validation still resulted in large uncertainties in the simulations of these
three variables. Therefore, future studies are still required for further validation of the model
performance using comprehensive observations covering these variables as well as the others, thus
reducing the simulation errors of the constraint and decision variables so as to improve the screening
precision of BMP alternatives.
**4.2 Biogeochemical effects of rotation pattern and other management practices**
The scenario analysis relying on model simulations in this study showed that environmental
nitrogen contamination could be reduced while i) sustaining crop yields to protect food security, ii)
achieving the 4‰ goal in soil carbon sequestration, and iii) decreasing the net ecosystem aggregate
GHG emission to mitigate climate change. The reductions in environmental nitrogen contamination
could be attributed to the better synchronization of crop nitrogen requirements and soil nitrogen
availability through optimizing field management practices.
For cotton, a period of 5 consecutive years is usually applied as the longest cotton monoculture to
stabilize its yields. Within this period, balanced elemental nutrients have been applied, and thus the
negative effect of monoculture on cotton yields can be offset in practice (Han, 2010). In addition, the
DNDC model assumes balanced nutrient supplies for any crops as well as optimum phytosanitary
conditions, and thus the negative effects of monoculture are not taken into account (e.g., Li, 2017).
The simulated positive annual $\triangle$SOC for the W-M cropping system were mainly attributed to the
incorporation of the full aboveground residues (at rates of 5.1−7.0 Mg C $ha^{-1}$ $yr^{-1}$), which in turn
favored for carbon sequestration (Han et al., 2016). On the contrary, the simulations of annual $\triangle$SOC
for the cotton cropping system were negative. The SOC stock decreases resulted from (i) the more
notable $CO_2$ emissions over the longer fallow season and (ii) the lower rates of fully incorporated
residues (at rates of 2.5−3.1 Mg C $ha^{-1}$ $yr^{-1}$) than those of the W-M (Liu et al., 2019). As a remarkable
carbon sink, the W-M cropping system with the incorporation of the full crop residues even could
completely compensate for the SOC lost during the first cotton-planting year following the transition.
Thus, the simulated annual ΔSOC was generally positive during the first cotton cultivation year of a
three-crop rotation cycle. As a result, the $R_0$ (i.e., pure W-M continuously within each 6-year period)
acted as a net GHG sink since the positive ΔSOC could exceed the $N_2O$ emission during the W-M
cultivation, whereas all the three-crop systems subject to $R_1$ to $R_5$ rotation patterns would function as
net GHG sources. The higher nitrogen application rate for the W-M than for the cotton resulted in more
reactive nitrogen remaining in the soil (Chen et al., 2014; Ju et al., 2009), thereby stimulating higher
emissions of $N_2O$ and nitrogenous air pollutants in the trials with fewer cotton cultivation years.
Therefore, the rotation patterns of the cotton and W-M can be optimized to realize sustainable
intensification in terms of sustaining crop yields at a relatively high level, maximizing SOC increase
and minimizing negative impacts on the climate and environment.

Northern China, as the most important agricultural region, experienced an increase in crop yields

by a factor of 2.8 from 1980 to 2008. During this period, the application of mineral fertilizers increased
by a factor of 5.1. The rapid increase in fertilizer use has resulted in excessive nitrogen remaining in the
soil, posing potential risks for the environment (Chen et al., 2011; Zhang et al., 2017). To solve this
problem, a reduction in fertilizer application was proposed in several previous studies (e.g., Chen et al.,
2011, 2014; Liu et al., 2012). The results of the scenario analysis in this study indicated that further
reducing the farmer-optimized nitrogen doses by 18% could still sustain the crop yields while greatly
decreasing the release of nitrogenous pollutants.

In addition to fertilization, over-irrigation has also been ubiquitous in northern China for a long

time, and is threatening the water security of this region due to the sharply declining groundwater table
and water pollution (Gao et al., 2015; Ju et al., 2009). For this reason, only management options that
can reduce the amount of irrigation water should be recommended due to the severe shortage of water
resources in the region. In addition, adopting sprinkling irrigation instead of flood irrigation for an
equal amount of water showed positive effects on the crop yields, indicating improved irrigation
efficiency (Zhang et al., 2017). This result indicated that increasing the water-use efficiency through
the application of alternative irrigation techniques in coupling with reduced nitrogen doses could be a
pathway to sustain crop yields.

587 Reduced tillage practices have been promoted in China in the recent years. To facilitate the

588 decomposition of the woody cotton residues and avoid outbreaks of diseases and pests induced by

589 continuous implementation of reduced tillage or no-till, the tillage practices were only adjusted in the

590 W-M fields, while currently applied deep tillage was maintained for the cotton when setting the tillage

591 scenarios. The simulations showed that the reduced tillage or no-till practices could sustain the crop

592 yields while reducing the $NH_3$ volatilization and $NO_3^-$ leaching, which were consistent with the reports

593 from experimental studies (e.g., Zhao et al., 2016).

594 As shown above, appropriate combinations of a rotation pattern and field management practices

595 can satisfy the three given constraints while resulting in the lowest NIPs with overlapping uncertainties.

596 However, direct observations in field experiments usually with very limited management treatments are

597 far less sufficient for screening these appropriate combination alternatives. However, identifying the

598 appropriate combination alternatives is one of the purposes of biogeochemical models, such as DNDC.

599 In principle, a biogeochemical model that is validated with limited observations from field experiments,

600 like the DNDC95 modified and used in this study, could be capable of fulfilling this task.

601 **4.3 Evaluation of the best management practice**

602 The scenario analysis in this study was effective for screening the BMP alternatives. The

603 identified BMP alternatives could sustain the crop yields of the three-crop rotation system, increase the

604 SOC stock annually at 4‰ for more, mitigate the NEGE, and reduce the $NH_3$ and NO emissions and

605 $NO_3^-$ leaching due to the enhanced resource use efficiency in response to the reduced nitrogen-fertilizer

606 doses, irrigation water amounts and tillage depth for the W-M. Hence, the BMP alternatives could

607 result in significantly reduced NIPs even compared to the currently applied field management practices

608 that have been optimized by the local farmers. However, the identified BMP alternatives were based on

609 the constraint and decision variables validated against only the observations at the single field site

610 involved in this case study. In this regard, confirmation of these BMP alternatives at other sites of this

611 region is still required in the future studies.

612 A biogeochemical model as an ideal tool for identifying the BMPs is reflected by near-zero $\varepsilon_s$ for

613 any constraint/decision variable or NIP. A small sample size of the observations used for validation of

614 any constraint/decision variable would result in largely positive or negative $\varepsilon_s$ (including large over- or

615 under-estimations) for model simulations of the variable, likely account for the large $\varepsilon_s$ of a NIP, and

thus lead to a lower precision in screening the BMP alternatives. Therefore, the applicability of the
approach proposed in this study for identifying the BMPs is highly dependent upon the validations
using observations with appropriate sample sizes for individual constraint and decision variables. In
this study, the $\varepsilon_s$ and $\varepsilon_{input}$ for the simulated variables and NIPs of management scenarios were
quantified. For the NIPs of the identified BMP alternatives, for instance, the $\varepsilon_s$ and $\varepsilon_{input}$ at relative
magnitudes were $6.5 \pm 4.9\%$ and $\pm 3.3\%$, respectively, which were similar with those ($9.1 \pm 5.0\%$ and
$\pm 3.1\%$, respectively) of baseline scenario. According to these errors, the uncertain ranges of the NIPs
for the three alternatives almost fully overlapped with each other while they were all beyond the
uncertain range of the NIP for the baseline scenario. This implicated that the approach proposed in this
study could be applicable for identifying the BMPs of the three-crop rotation system. Nevertheless, the
$\varepsilon_s$ uncertain range of one times SD still fully diverged negatively from zero, due to the marginally small
sample sizes of available $\triangle$SOC, $NH_3$ volatilization and $NO_3^-$ leaching observations that led to
insufficient validations for these variables. Especially, the model underestimations of $NO_3^-$ leaching
(with an adjusting factor of 1.42 and error factors of $-29 \pm 4\%$) overwhelmingly dominated the
diverged $\varepsilon_s$ of the NIPs, which were comparable or larger than the $\varepsilon_{input}$ values. Relying on the few field
observations, one was still not able to judge whether there are insufficiencies in the scientific structures
or inappropriate parameters in the model to dominate the large $\varepsilon_s$ for these variables. Therefore,
multiple comprehensive field observations with appropriate sample sizes to fully cover all the relevant
variables are as substantially necessary as an advanced biogeochemical model with multiple functions
in order to address the best management issue of a multi-crop rotation system to achieve multiple
benefits.

The DNDC model has been established by following the mass conservation law. In other words,

this model can accurately reflect the mass balance of the carbon or nitrogen budgets for the simulated
soil layer (0−50 cm depth). This principle implies that only one nitrogen budget item could be omitted
for model validation. This item is usually soil nitrogen loss through the production of dinitrogen gas
($N_2$), mainly by denitrification, which is very difficult to measure *in situ* (e.g., Wang et al., 2013; Zhang
et al., 2019). For both crop cropping systems, however, the nitrogen lost through this pathway could be
almost fully inhibited in the topsoil, wherein the soil moisture contents were often lower than 60%
WFPS (Linn and Doran, 1984; Liu et al., 2011, 2014). For instance, the $N_2$ emission likely accounts for
approximately 1.6% of the urea applied in a winter wheat season (Zhang et al., 2019), which is at a
negligibly low level for the nitrogen balance.

Regarding the identification of the BMPs, the approach proposed and applied in this case study

only includes the biogeochemical effects of management on the constraint/decision variables. This
approach currently excludes other factors, such as those related to the costs of the management
practices, thereby likely resulting in uncertainties in the screened BMP alternatives. Despite some
deficiencies, this approach can be easily and automatically implemented as long as the simulations for
all constraint and decision variables can be validated using comprehensive observations, implicating its
potential applicability for more comprehensive situations. Adding the missing factors is one of the
future research tasks to further improve this approach.
**5 Conclusions**

To address the challenging issue for optimizing multi-crop system management to simultaneously

achieve multiple benefits, a biogeochemical model-based approach for identifying the best
management practices (BMPs) was proposed and tested in this site-scale case study. A three-crop
system widely distributed in northern China, which grew cotton in rotation with winter wheat and
summer maize (W-M), was investigated. The BMPs were referred to the management alternatives with
the lowest negative impact potentials (NIPs), falling overlapping uncertain ranges, among the scenarios
satisfying a set of constraints. The NIP of a scenario was defined as the linear function of five decision
variables, including the net ecosystem aggregate greenhouse gas emission (NEGE), ammonia ($NH_3$)
volatilization, nitric oxide (NO) release, emission of nitrous oxide ($N_2O$) as an ozone layer depletion
matter, and nitrate leaching. This study used three variables, i.e., crop yield, annual change in SOC
stock ($\triangle$SOC), and NEGE, to specify the applied constraints that were stable/increased crop yields,
annual $\triangle$SOC by 4‰ or more, and reduced annual NEGE by at least 5% in comparison with those of
the baseline scenario (as the currently applied practices in this study). The constraint and decision
variables to determine the NIP of each scenario were provided by the simulation of
DeNitrification-DeComposition version 95 model (DNDC95) modified in this study. Due to the
unsatisfactory performance of the model in daily simulations of NO emission and net ecosystem
exchange of carbon dioxide (NEE), the model was modified to include a new parameterization of soil
moisture effects on the NO production during nitrification and replacement of the original calculation
approach for NEE with an algorithm based on gross primary production. For the concerned variables
with available measurements in two adjacent lands at the selected field site, the modified model
showed statistically meaningful consistence between simulations and observations. Using the
systematic errors obtained from the model validation to determine the simulation uncertainties of the
concerned variables for each scenario and that of its NIP, the modified model simulations driven by
6000 management scenarios automatically identified three BMP alternatives. These BMP alternatives
follow the current adopted rotation pattern (3 consecutive years of cotton rotated with 3 continuous
years of W-M) applied with 18% less fertilizer nitrogen and ~23% less irrigation water through
sprinkling or flooding and reduced depth of tillage for the W-M even in comparison with the current
applied farmer-optimized management practices. This case study demonstrated the practicability of the
model-based approach and implicated its potential applicability for optimizing the field management of
multi-crop system to simultaneously achieve multiple United Nations Sustainable Development Goals.
It also emphasized the need to make comprehensive observations that fully cover the constraint and
decision variables, other related factors as well as all the crops and management practices in question to
facilitate effective BMP screening through virtual experiments using a biogeochemical model, such as
DNDC. In the future study to identify the BMPs specifically for the three-crop rotation system at the
regional scale, it is still necessary for a 6-year model validation that includes a rotation of all three
commodity crops as well as all studied management practices in question.
**Data availability**
All the model output for producing the figures can be obtained from the supplementary materials and
all the observed data sets used in this study can be available from the co-authors.
**Author contributions**
X, Zheng, C, Liu and J, Zhu contributed to develop the idea and enhance the science of this study. W,
Zhang proposed a new evaluation factor – negative impact potential, designed and implemented the
model simulations and virtual experiments and prepared the manuscript with contributions from all
co-authors. C, Liu, K, Wang, R, Wang and Z, Yao contributed to obtain the field measured data. F, Cui
and S, Li contributed to the model validation for the winter wheat-summer maize cropping system.
**Competing interests**
The authors declare that they have no conflict of interest.
**Acknowledgement**
This study was jointly supported by the National Key R&D Program of China (2016YFA0602303) and
the National Natural Science Foundation of China (41603075, 41761144054).

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

Butterbach-Bahl, K.: Quantifying net ecosystem carbon dioxide exchange of a short-plant
cropland with intermittent chamber measurements, Global Biogeochem. Cycl., 22, GB3031.

 **Table and figure captions**

Table 1 Simulated constraint and decision variables and negative impact potentials (NIPs) for the
baseline (the conventionally applied practices) and the alternatives of the best management practices.
Figure 1: Observed and simulated daily mean soil (5 cm) temperature, soil (0−6 cm) moisture, daily net
ecosystem exchanges of carbon dioxide (NEE) in cotton field and winter wheat-summer maize fields,
and daily fluxes of methane ($CH_4$), nitrous oxide ($N_2O$) and nitric oxide (NO) from cotton field. The
solid- and dashed-line arrows indicate the dates of fertilization and irrigation, respectively. The
measurement errors were not shown in panels a−e for figure clarity. The vertical bar for each
observation in panels f−h indicates double standard deviations to represent the uncertain at the 95%
confidence interval. The legends in panel c apply for all subfigures.
Figure 2: Comparison between observations and simulations of crop yields, annual/seasonal cumulative
NEE and NEGE, and annual/seasonal ΔSOC, and annual cumulative fluxes of methane ($CH_4$) uptake,
nitrous oxide ($N_2O$) and nitric oxide (NO), and cumulative fluxes of ammonia ($NH_3$). Yield, seed yield
of cotton (open cycle) and grain yield of winter wheat (solid cycle) and summer maize (solid diamond).
NEE, net ecosystem exchanges of carbon dioxide. NEGE, net ecosystem aggregate greenhouse gas
emission. ΔSOC, change in soil organic carbon stock. Given NEE, NEGE and ΔSOC are annual for
cotton and seasonal for wheat and maize. The observed ΔSOC was given as the opposite of NEE plus
yield in carbon mass quantity for the cropping system with incorporation of full residues whereas each
ΔSOC simulation was the sum of simulated changes in carbon stocks of soil humus, microbial biomass
and dissolvable organic compounds. Simulations were resulted from the modified model. Given slope
errors of the zero-intercept linear regressions are double standard deviations to represent the 95%
confidence interval. Vertical bars indicate standard deviation of three or four spatial replicates, with
exception for NEE. Given errors of NEE were adapted from the coefficient of variation on average
(25%) reported by Wang et al. (2013b). DM, dry matter. $CO_2$eq, carbon dioxide equivalent. The
100-year global warming potentials of 34 for $CH_4$ and 298 for $N_2O$ (IPCC, 2013) were used to quantify
NEGE in $CO_2$eq quantity.
Figure 3: Simulated cumulative crop yields, changes in soil organic carbon (ΔSOC), methane ($CH_4$),
nitrous oxide ($N_2O$) releases, net ecosystem aggregate greenhouse gas emission (NEGE), ammonia
($NH_3$) volatilization, nitric oxide (NO) emission and nitrate leaching (NL) of individual rotation
patterns (with a 6-year rotation cycle) over a 18-year period. $R_0$, $R_1$, ..., $R_5$ represents the rotation
pattern with the cotton cultivated consecutively for 0, 1, ..., 5 year(s), respectively, within each 6-year
rotation cycle. The legends in panel e apply for all subfigures. Given simulations resulted from the
modified model driven by the currently applied field management practices (i.e., the baseline field
management scenario) and observed means of input soil properties.
Figure 4: Simulated effects of various rotation patterns between cotton and winter wheat-summer
maize cropping system with a 6-year cycle on decision variables and negative impact potential (NIP).
The subscript of $R_0$, $R_1$, ..., $R_5$ are referred to the number of consecutive years for cotton cultivation.
The y-axis units are Mg C ha$^{-1}$ yr$^{-1}$ for the opposite of mean annual increase in soil organic carbon
stock ($-\Delta$SOC), kg C ha$^{-1}$ yr$^{-1}$ for methane ($CH_4$) emission, kg N ha$^{-1}$ yr$^{-1}$ for fluxes of nitrous oxide
($N_2O$), ammonia ($NH_3$) and nitrous oxide (NO), and nitrate leaching (NL), Mg $CO_2$eq ha$^{-1}$ yr$^{-1}$ for net
ecosystem aggregate greenhouse gas emission (NEGE), and US$ ha$^{-1}$ yr$^{-1}$ for NIP. The $CO_2$eq was
based on the 100-year global warming potentials, i.e., 34 for $CH_4$ and 298 for $N_2O$ (IPCC, 2013). The
NIP was calculated using Eq. 7 presented in the text. The vertical bar within the open cycle of each
datum point indicates the absolute uncertainty (1 standard deviation) induced by input uncertainties of
key soil properties. Each unfilled column indicates the absolute total uncertainty of the simulation, with
its vertical bar representing its random uncertainty (1 standard deviation).
Table 1 Simulated constraint and decision variables and negative impact potentials (NIPs) for the baseline (the conventionally applied practices) and the alternatives of the
best management practices.

| Scenarios | | | | | | R | Constraint variable | | | | | | | | | | | Decision variable | | | | | NIP | | |
|---|---|---|---|---|---|---|---|---|---|---|---|---|---|---|---|---|---|---|---|---|---|---|---|---|---|
| | | | | | | | Yield | | | | | ΔSOC | | | NEGE | | | | | | | | | | | |
| | | N | IA | IM | T | | $Sim_{cotton}$ | $Sim_{wheat}$ | $Sim_{maize}$ | $\varepsilon_s$ | $\varepsilon_{input}$ | Sim | $\varepsilon_s$ | $\varepsilon_{input}$ | Sim | $\varepsilon_s$ | $\varepsilon_{input}$ | $CH_4$ | $N_2O$ | $NH_3$ | NO | NL | Sim | $\varepsilon_s$ | $\varepsilon_{input}$ |
| BAS | 110/430 | 100 | IF | 20 | $R_3$ | | 3.5 | 4.8 | 6.8 | 0.15 (0.08) | 0.04 | 0.14 | 0.03 (0.02) | 0.02 | 1.06 | 0.22 (0.15) | 0.18 | −1.88 | 3.55 | 57 | 1.60 | 58 | 453 | −41 (22) | 14 |
| $BMP_1$ | 90/353 | 79 | IS | 5 | $R_3$ | | 3.6 | 4.8 | 6.8 | 0.15 (0.08) | 0.03 | 0.19 | 0.04 (0.03) | 0.02 | 0.98 | 0.20 (0.14) | 0.19 | −1.81 | 3.71 | 43 | 1.57 | 33 | 332 | −22 (16) | 11 |
| $BMP_2$ | 90/353 | 76 | IS | 5 | $R_3$ | | 3.6 | 4.8 | 6.8 | 0.15 (0.08) | 0.03 | 0.18 | 0.04 (0.03) | 0.02 | 0.98 | 0.20 (0.14) | 0.20 | −1.81 | 3.71 | 43 | 1.57 | 33 | 333 | −22 (16) | 11 |
| $BMP_3$ | 90/353 | 76 | IF | 5 | $R_3$ | | 3.5 | 4.8 | 6.8 | 0.15 (0.08) | 0.03 | 0.18 | 0.04 (0.03) | 0.01 | 1.00 | 0.21 (0.15) | 0.20 | −1.83 | 3.76 | 44 | 1.56 | 33 | 335 | −22 (16) | 11 |

[a] BAS, the baseline. BMP, different best management alternatives denoted by subscript numbers. N, nitrogen fertilizer dose (kg N ha$^{-1}$ yr$^{-1}$) of cotton/wheat-maize (W-M). IA, irrigation water
amount (mm per event). IM, irrigation method. IF, flood-irrigation. IS, sprinkling irrigation. T, tillage depth (cm). R, rotation pattern. $R_3$, rotation pattern with 3 consecutive years of cotton
rotated with 3 continuous years of W-M. Yield, seed (cotton) or grain (W-M) yield (Mg ha$^{-1}$ in dry matter). Sim, annual quantity simulation. $\varepsilon_s$, the absolute total simulation error (i.e., the
systematic error) of the annual quantity simulation, with its error (1 standard deviation) representing the random uncertain magnitude. $\varepsilon_{input}$, the random model simulation error (1 standard
deviation) due to input uncertainties of the key soil properties including clay fraction, bulk density, pH and soil organic carbon content. ΔSOC, annual change in soil organic carbon stock in the
0−50 cm (Mg C ha$^{-1}$ yr$^{-1}$). NEGE, net ecosystem aggregate greenhouse gas emission in carbon dioxide equivalent (Mg $CO_2$eq ha$^{-1}$ yr$^{-1}$). $CH_4$, methane emission (kg C ha$^{-1}$ yr$^{-1}$). $N_2O$, $NH_3$,
NO and NL, emission of nitrous oxide, ammonia, and nitric oxide, and nitrate leaching, respectively (kg N ha$^{-1}$ yr$^{-1}$). The $CO_2$eq was based on the 100-year global warming potentials of 34 for
$CH_4$ and 298 for $N_2O$ (IPCC, 2013). NIP, negative impact potential (US\$ ha$^{-1}$ yr$^{-1}$). Given simulations are the averages of 18 consecutive years.

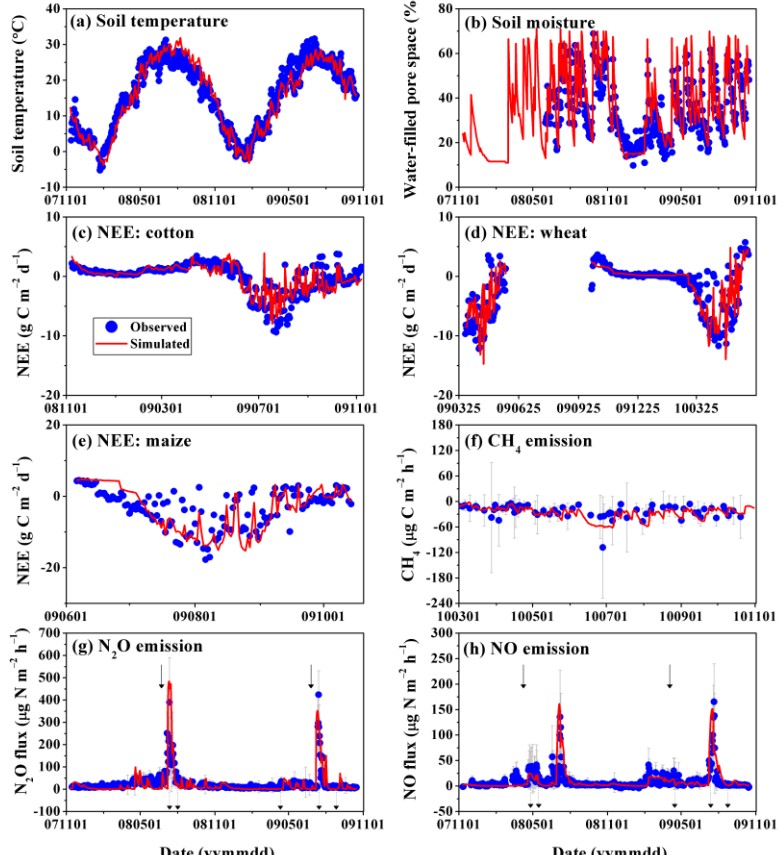

Figure 1: Observed and simulated daily mean soil (5 cm) temperature, soil (0−6 cm) moisture, daily net ecosystem exchanges of carbon dioxide (NEE) in cotton field and winter wheat-summer maize fields, and daily fluxes of methane ($CH_4$), nitrous oxide ($N_2O$) and nitric oxide (NO) from cotton field. The solid- and dashed-line arrows indicate the dates of fertilization and irrigation, respectively. The measurement errors were not shown in panels a−e for figure clarity. The vertical bar for each observation in panels f−h indicates double standard deviations to represent the uncertain at the 95% confidence interval. The legends in panel c apply for all subfigures.

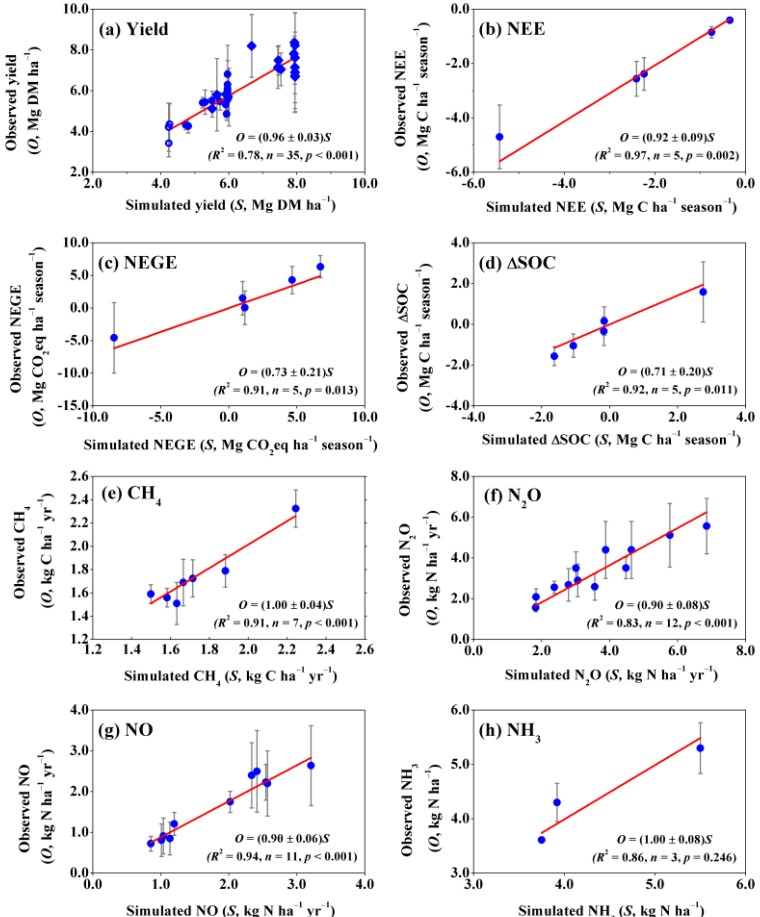

933

**Figure 2: Comparison between observations and simulations of crop yields, annual/seasonal cumulative NEE and NEGE, and annual/seasonal ∆SOC, and annual cumulative fluxes of methane (CH₄) uptake, nitrous oxide (N₂O) and nitric oxide (NO), and cumulative fluxes of ammonia (NH₃). Yield, seed yield of cotton (open cycle) and grain yield of winter wheat (solid cycle) and summer maize (solid diamond). NEE, net ecosystem exchanges of carbon dioxide. NEGE, net ecosystem aggregate greenhouse gas emission. ∆SOC, change in soil organic carbon stock. Given NEE, NEGE and ∆SOC are annual for cotton and seasonal for wheat and maize. The observed ∆SOC was given as the opposite of NEE plus yield in carbon mass quantity for the cropping system with incorporation of full residues whereas each ∆SOC simulation was the sum of simulated changes in carbon stocks of soil humus, microbial biomass and dissolvable organic compounds. Simulations were resulted from the modified model. Given slope errors of the zero-intercept linear regressions are double standard deviations to represent the 95% confidence interval. Vertical bars indicate standard deviation of three or four spatial replicates, with exception for NEE. Given errors of NEE were adapted from the coefficient of variation on average (25%) reported by Wang et al. (2013b). DM, dry matter. CO₂eq, carbon dioxide equivalent. The 100-year global warming potentials of 34 for CH₄ and 298 for N₂O (IPCC, 2013) were used to quantify NEGE in CO₂eq quantity.**

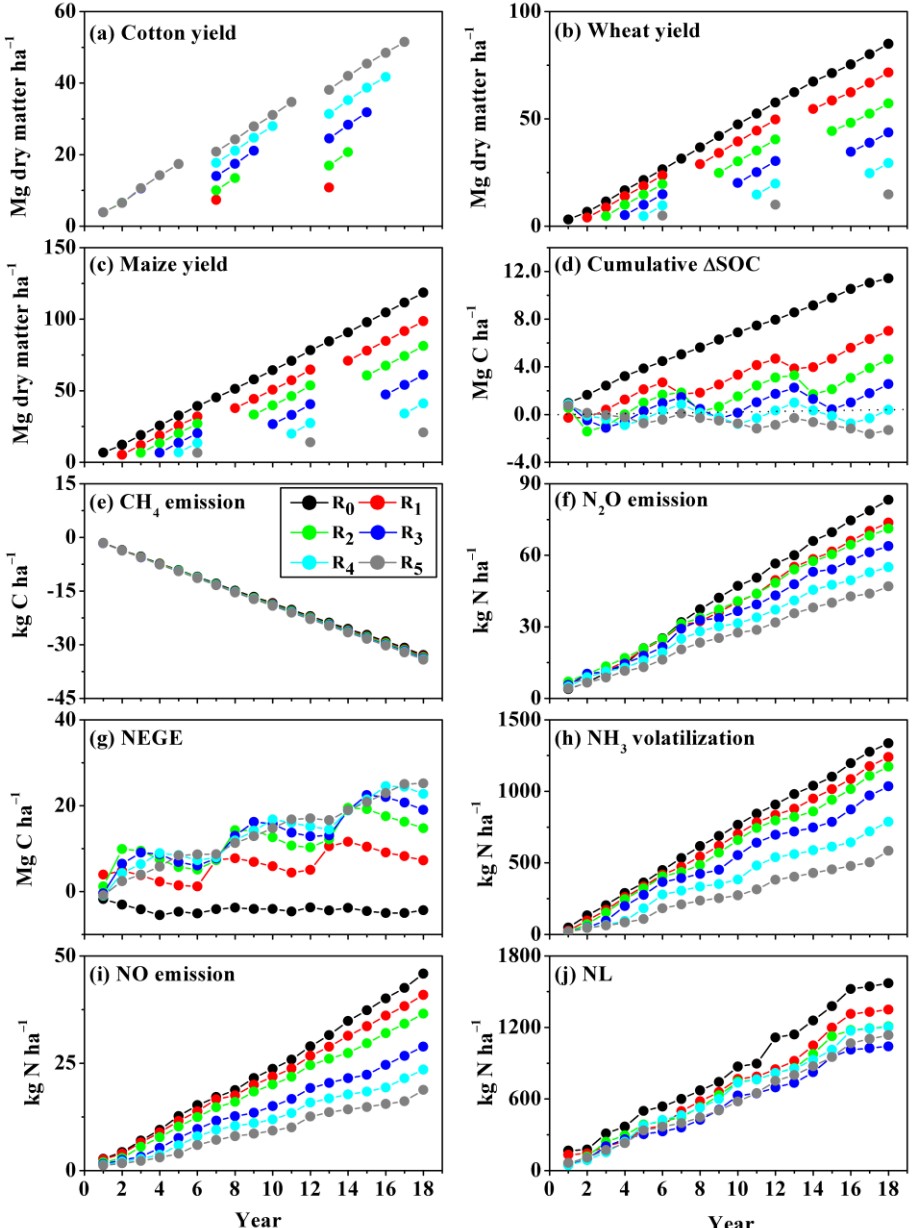

949

**Figure 3: Simulated cumulative crop yields, changes in soil organic carbon (ΔSOC), methane (CH₄), nitrous**

**oxide (N₂O) releases, net ecosystem aggregate greenhouse gas emission (NEGE), ammonia (NH₃)**

**volatilization, nitric oxide (NO) emission and nitrate leaching (NL) of individual rotation patterns (with a**

**6-year rotation cycle) over a 18-year period. $R_0$, $R_1$, ..., $R_5$ represents the rotation pattern with the cotton**

**cultivated consecutively for 0, 1, ..., 5 year(s), respectively, within each 6-year rotation cycle. The legends in**

**panel e apply for all subfigures. Given simulations resulted from the modified model driven by the currently**

**applied field management practices (i.e., the baseline field management scenario) and observed means of**

**input soil properties.**

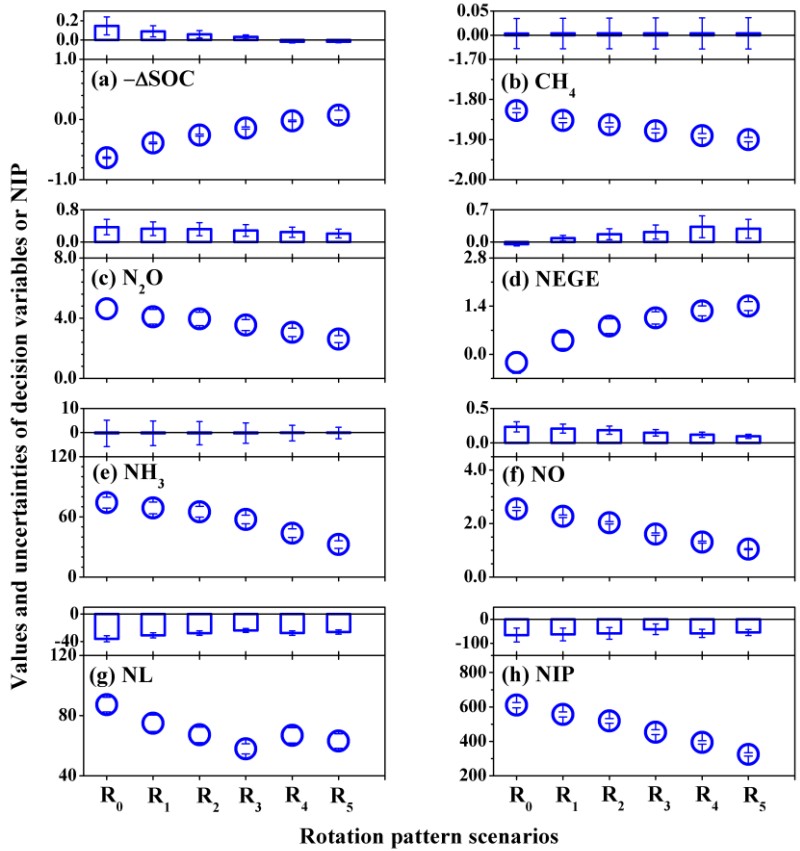

958

**Figure 4: Simulated effects of various rotation patterns between cotton and winter wheat-summer maize cropping system with a 6-year cycle on decision variables and negative impact potential (NIP). The subscript of $R_0$, $R_1$, ..., $R_5$ are referred to the number of consecutive years for cotton cultivation. The y-axis units are Mg C ha$^{-1}$ yr$^{-1}$ for the opposite of mean annual increase in soil organic carbon stock ($-\Delta$SOC), kg C ha$^{-1}$ yr$^{-1}$ for methane (CH$_4$) emission, kg N ha$^{-1}$ yr$^{-1}$ for fluxes of nitrous oxide (N$_2$O), ammonia (NH$_3$) and nitrous oxide (NO), and nitrate leaching (NL), Mg CO$_2$eq ha$^{-1}$ yr$^{-1}$ for net ecosystem aggregate greenhouse gas emission (NEGE), and US\$ ha$^{-1}$ yr$^{-1}$ for NIP. The CO$_2$eq was based on the 100-year global warming potentials, i.e., 34 for CH$_4$ and 298 for N$_2$O (IPCC, 2013). The NIP was calculated using Eq. 7 presented in the text. The vertical bar within the open cycle of each datum point indicates the absolute uncertainty (1 standard deviation) induced by input uncertainties of key soil properties. Each unfilled column indicates the absolute total uncertainty of the simulation, with its vertical bar representing its random uncertainty (1 standard deviation).**