# Peer review of "Using modified DNDC biogeochemical model to optimize"

_Biogeosciences, 2018_

## Referee Comment (RC1) · Anonymous Referee #1 · 5 Dec 2018

The manuscript entitled "Modeling the biogeochemical effects of rotation pattern and field management practices in a multi-crop (cotton, wheat, maize) rotation system: a case study in northern China" is within the scope of BG. To ensure reliability, models should be tested and improved as part of their development and application. The manuscript is important in that context (though it is poor- it lacks for a 6 years validation that includes a rotation of all three commodity crops as well as all management practices studied in question) but the novelty of this manuscript lies with the optimization of different rotation patterns (of three cultivars: cotton, wheat, maize) and management

practices which is very complex. Overall, the manuscript lacks of structure and the English language in the manuscript needs to be improved. The manuscripts need major revisions to be acceptable for the publication. In the current state should be rejected. For more details please see my comments below:

In the site simulation NEE and NO emission are predicted with lower accuracy by the model, then how this impacted the optimization of mitigation options?

The novelty of this manuscript lies with optimization of mitigation options at site level but authors exploited this inadequately in this manuscript. Elaborating and extending optimization analysis will add substantial knowledge and value to the manuscript. What about using i.e. Monte Carlo optimization technique to screen different set of possible agricultural management practices (a multiple optimization criteria that includes crop rotation in interaction with all studied management practices) which maximize yields while minimizing environmental effects.

Uncertainty quantification is a critical challenge in both validation and calibration. There is NO mention of model uncertainty in the manuscript. I suggest adding one section on model uncertainty and discussing uncertainties and how that might propagate to model outputs in this study. Authors should also focus on potential applications of optimization considering uncertainty. Otherwise these mitigation options have only academic interest and not much real-world value. Please, see the specific comments below.

Introduction: In general I would say that the introduction is too long and not enough focused on the task. There are plenty of paragraphs which must be shortened and better structured. This will improve the content and impact of the current manuscript. Please skip unnecessary things. i.e. frequent applications of pesticides and/or herbicides. My suggestion is to reduce the introduction section to max. 2 pages.

I will start with one example: Globally, fiber crops (i.e. cotton) and cereals such as wheat and maize have been playing a relevant role in humanity as they are a primary source for the textile and food industry. In China, while the cultivation of cotton only

covers between 2.0−3.9% of the annual crop harvest areas (cotton lint production of 5.3−7.6 million metric tons during 2007−2016), the cultivation of cereals is significantly large. Wheat and maize account for 39% and 26% of the harvest area and represent 129 and 220 million metric tons of grain in 2016, respectively (China Statistical Yearbook, 2017).

Northern China is not only the second most important area of cotton production but the largest region of the winter wheat−summer maize double-cropping system (i.e., both crops harvested within a year, hereinafter referred to as W-M) in the country (e.g., Cui et al., 2014). Crop rotations of cotton and the W-M have been commonly applied in this region (e.g., Liu et al., 2010, 2014) and are typically alternated every 3−5 years. During the last decades, cotton, wheat and maiz's yields have increased by means of intensified agricultural management practices such as: increased fertilizer inputs, advanced irrigation methods (Han, 2010). A recent study (Liu et al., 2019) indicated that the cotton cropping system in northern China persistently functioned as an intensive carbon or net greenhouse gas (GHG) source compared to the W-M because of strong carbon dioxide ($CO_2$) emissions during the long non-growing periods. Add Reference.. revealed that the change in storage of soil organic carbon (âŰşSOC), net ecosystem GHG emission (NEGE) and other biogeochemical processes of the multiple-cropping systems in northern China likely are closely related to the rotation pattern of cotton and the W-M. Thus, one can hypothesize that identifying and adopting optimal rotation pattern of cotton and the W-M are beneficial for soil carbon sequestration and mitigation of GHG emissions in the region. . . .. . . . Please see general comment of this section!

Material and Methods

General comment: Same as above, please shorten and restructure this section

Put sections 2.1 and 2.3 together (short and concise)

Lines 222-226 what do you want to say? It is not clear to me. Please keep in mind that you are not studying the environmental impacts of using pesticides.

Discussion

Please delete lines 526-527. I do not see that such statement helps to your work. Unfortunately – your model validation is poor as it evaluates only one site and does not include a rotation of all three commodity crops together. Remember that optimization studies rely on robust site validations. These validation studies should be done using several pilot areas with different geographical, climatic and soil conditions; different types of reference data (long term datasets) used for model calibration. I am not sure that you will get the same results if you apply your best rotation and management practices across different geographical, climatic and soil conditions. A regional simulation will help you to clarify this.

I would start with this: The scenario analysis relying on model simulations in this study showed that environmental contamination can be reduced while a) sustaining crop yields and b) increasing carbon sequestration in the soil. Reductions of environmental i.e. N losses are attributed to the better synchronization of crop N requirements and soil N availability.........

Lines 531-532 Why do you discuss about pesticides when the DNDC model does not account for?. Please state that DNDC model assumes balanced nutrient supplies for any crops as well as optimum phytosanitary conditions. Hence negative effects of monoculture are not accounted for.

I suggest you to add an uncertainty section as requested before.

---

## Referee Comment (RC2) · Anonymous Referee #2 · 28 Dec 2018

This manuscript describes a modeling effort to evaluate the biogeochemical effects of optimizing a cotton, wheat, maize rotation and field management practices. This work is within the scope of Biogeosciences. Modeling efforts such as this are important given the difficulty in designing field experiments that adequately capture the appropriate biogeochemical parameters for each treatment. Overall this work is important in increasing our understanding the environmental impacts of management practices. However, one major weakness is a lack of sufficient experimental validation.

Specific comments Page 3, lines 67-68 describe the "release potentials of nitrogenous

pollutants". It would be helpful to describe these pollutants in the context of agricultural practices (e.g. nitrate leaching, etc.).

Page 6, Section 2.2: This description of the DNDC95 model is somewhat redundant with the introduction and could either be shortened here or removed from the introduction.

Page 8, Section 2.4: Six level-I scenarios are described with increasing the number of cotton rotations with W-M. How were multiple cotton crops incorporated? For R2 was it 2 years of cotton followed by 4 years W-M or 1 or 2 years of W-M between each cotton crop?

Page 23, Section 4.3: This discussion of the BMP should also include a discussion of the potential impacts of weed or disease pressure. Continuous cultivation of these crops could lead to challenges for weed or disease management which does not appear to be adequately addressed by the model. This section should include a discussion of these limitations.

Page 25: The number of scenarios mentioned in the discussion is different from page 9 and 18.

Figure 2: It is difficult to differentiate between the different rotation patterns since most of the symbols are stacked. An additional figure or table showing the order of rotations for R0-R5 would be helpful.

Technical corrections Line 79: Remove "s" from "contents" Line 132: Replace "its" with "associated" Line 619 should not be indented

---

## Author Comment (AC1) · 16 Jan 2019

RC1 The manuscript entitled "Modeling the biogeochemical effects of rotation pattern and field management practices in a multi-crop (cotton, wheat, maize) rotation system: a case study in northern China" is within the scope of BG. To ensure reliability, models should be tested and improved as part of their development and application. The manuscript is important in that context (though it is poor- it lacks for a 6 years validation that includes a rotation of all three commodity crops as well as all management practices studied in question) but the novelty of this manuscript lies with the optimization of

different rotation patterns (of three cultivars: cotton, wheat, maize) and management practices which is very complex. Overall, the manuscript lacks of structure and the English language in the manuscript needs to be improved. The manuscripts need major revisions to be acceptable for the publication. In the current state should be rejected.

Revised. We have revised the manuscript one by one as the reviewer suggested. The manuscript has been new structured, especially for the Introduction, and the English language has been revised by AJE. Please see the detailed revision notes below.

For more details please see my comments below: In the site simulation NEE and NO emission are predicted with lower accuracy by the model, then how this impacted the optimization of mitigation options?

Revised. "The simulated NEE flux is one of component of âŰşSOC, which is key factor considered in BMP selection." (Please see lines 233-234). âŰşSOC and NO are all the decision variables for NIP, which is applied for screening the BMP. "the NEGE was the residual of the annual sum of CH4 and N2O emissions minus the âŰşSOC" and "NEGE, NH3, NO, N2OODM, and NL represent the multi-goal decision variables" (Please see lines 183-185 and lines 189-192). The relative uncertainty of the model validation was evaluated, and then we provided the uncertainty of the BMP, which reflect the impacts of validation on BMP. "In this study, the MRB of simulated variables were regarded as the relative uncertainty of the model validation, which were further used for estimating the relative uncertainty of each scenario based on the error transfer formula (Eqs. (S1-4))." (Please see lines 210-212) and section "3.4 The uncertainty of the best management practice". (Please see lines 309-321)

The novelty of this manuscript lies with optimization of mitigation options at site level but authors exploited this inadequately in this manuscript. Elaborating and extending optimization analysis will add substantial knowledge and value to the manuscript. What about using i.e. Monte Carlo optimization technique to screen different set of possible agricultural management practices (a multiple optimization criteria that includes crop

rotation in interaction with all studied management practices) which maximize yields while minimizing environmental effects.

Revised. The Monte Carlo technique has been applied for identifying the BMP. "To screen the BMP of six rotation patterns in interaction with all considered management practices, the variation of fertilizer amount, irrigation amount and residue incorporation rate was set as 40% of baseline to baseline (N44/172 to N110/430), 40% of baseline to baseline (I40 to I100) and 0 to 100% (RI0 to RI100). The factors of irrigated method and tillage consisted of flood (IF) and sprinkle (IS) irrigation, and no-tillage (T0), reduced tillage (5 cm and 10 cm, T5 and T10) and conventional tillage (20 cm, T20), respectively. We assumed the frequency distribution of all the factors were uniform. Monte Carlo simulations, 1000 combination scenarios of field managements, were used to screen the BMP for each rotation pattern, and the final BMP for the system were selected from the BMPs of six rotation patterns in light of 6000 combination scenarios." (Please see lines 161-170)

Uncertainty quantification is a critical challenge in both validation and calibration. There is NO mention of model uncertainty in the manuscript. I suggest adding one section on model uncertainty and discussing uncertainties and how that might propagate to model outputs in this study. Authors should also focus on potential applications of optimization considering uncertainty. Otherwise these mitigation options have only academic interest and not much real-world value. Please, see the specific comments below.

Revised. The section on model relative uncertainty of the validation was added, and then we provided the uncertainty of the BMP, which reflect the impacts of validation on BMP. "In this study, the MRB of simulated variables were regarded as the relative uncertainty of the model validation, which were further used for estimating the relative uncertainty of each scenario based on the error transfer formula (Eqs. (S1-4))." (Please see lines 210-212) and section 3.4 "The relative uncertainty resulted from the model validation was calculated based on MRB and error transfer formula (Eqs. (S1-4)). The MRB of cumulated $N_2O$, NO, NEE and $CH_4$ were 3%, 6%, 2% and 8%,

respectively, which were used for calculating the relative uncertainty of NIP for all 6000 scenarios. For the BMP of each rotation pattern, the scenarios, whose uncertainty ranges had some overlap as that of the BMP, showed no significant differences from each other. Thus, 7, 8, 4, 2 and 2 alternative scenarios were selected for the BMP of R0, R1, R2, R3 and R4, respectively, with the average relative uncertainty of 5.1%. For the final identified BMP of N90/353_I82_IS_RI90_T10 involved in R3 rotation pattern, the relative uncertainty of the NIP was 4.7%, ranged from 314 to 345 USD ha−1 yr−1. There were another two alternative scenarios (N94/366_I91_IS_RI95_T10 and N97/378_I88_IS_RI70_T5) in R3, which indicated the trade-off effects of different field managements, such as the opposite effect of reduced residue incorporation (decrease âŰşSOC) and tillage depth (increase âŰşSOC) on âŰşSOC. They were also regarded as alternative BMP for the system." (Please see lines 309-321)

Introduction: In general I would say that the introduction is too long and not enough focused on the task. There are plenty of paragraphs which must be shortened and better structured. This will improve the content and impact of the current manuscript. Please skip unnecessary things. i.e. frequent applications of pesticides and/or herbicides. My suggestion is to reduce the introduction section to max. 2 pages. I will start with one example: Globally, fiber crops (i.e. cotton) and cereals such as wheat and maize have been playing a relevant role in humanity as they are a primary source for the textile and food industry. In China, while the cultivation of cotton only covers between 2.0−3.9% of the annual crop harvest areas (cotton lint production of 5.3−7.6 million metric tons during 2007−2016), the cultivation of cereals is significantly large. Wheat and maize account for 39% and 26% of the harvest area and represent 129 and 220 million metric tons of grain in 2016, respectively (China Statistical Yearbook, 2017). Northern China is not only the second most important area of cotton production but the largest region of the winter wheat−summer maize double-cropping system (i.e., both crops harvested within a year, hereinafter referred to as W-M) in the country (e.g., Cui et al., 2014). Crop rotations of cotton and the W-M have been commonly applied in this region (e.g., Liu et al., 2010, 2014) and are typically alternated every 3−5 years.

During the last decades, cotton, wheat and maize's yields have increased by means of intensified agricultural management practices such as: increased fertilizer inputs, advanced irrigation methods (Han, 2010). A recent study (Liu et al., 2019) indicated that the cotton cropping system in northern China persistently functioned as an intensive carbon or net greenhouse gas (GHG) source compared to the W-M because of strong carbon dioxide ($CO_2$) emissions during the long non-growing periods. Add Reference. revealed that the change in storage of soil organic carbon (âŰşSOC), net ecosystem GHG emission (NEGE) and other biogeochemical processes of the multiple-cropping systems in northern China likely are closely related to the rotation pattern of cotton and the W-M. Thus, one can hypothesize that identifying and adopting optimal rotation pattern of cotton and the W-M are beneficial for soil carbon sequestration and mitigation of GHG emissions in the region...... Please see general comment of this section!

Revised. The Introduction has been structured and revised based on reviewer's comments, and the length has been reduced to less than 2 pages, with words reduction from 1400 to 800. For instance, "The objective method is applied for identifying the best management practice (BMP), which evaluates each decision variable with price-based proxies or other measures and screens the best option with minimal negative impact potential (NIP) under the given constraints (e.g., Cui et al., 2014; Xu et al., 2017). To screen the BMP using the multi-goal approach, it is essential to quantify the biogeochemical effects of management practices. As field experiments often focus only on the decision variables of very few management practices during short periods (e.g., Ding et al., 2007; Liu et al., 2010, 2015; Wang et al., 2013a, b), the process-oriented biogeochemical models have the potential to overcome this limitation, such as the DNDC (e.g., Chen et al., 2016; Giltrap et al., 2010; Li, 1992, 2000; Zhang et al., 2017a), DAYCENT (e.g., Delgrosso et al., 2005) and LandscapeDNDC (e.g., Haas et al., 2012; Molina-Herrera et al., 2016) models." (Please see lines 33-86)

Material and Methods General comment: Same as above, please shorten and restructure this section Put sections 2.1 and 2.3 together (short and concise)

[Figure]

Revised. We shorten and revised the content of Materials and methods as the reviewer suggested. Put the sections 2.1 and 2.3 together as section 2.2 and reconstructed. For instance, "Based on the model validation of Cui et al. (2014) for the W-M, the authors were able to further validate this model in the adjacent land cultivated with cotton. The daily meteorological data of 2004−2010 were directly obtained from Cui et al. (2014). Measured data were directly used for the least required soil properties. The input data of field capacity and wilting point in water-filled pore space (WFPS), being 0.65 and 0.2, respectively, were cited from Cui et al. (2014). The crop parameters for cotton were directly determined by the field measurements, which were 1900 kg C ha−1 for potential grain (1.2 times the mean of the measured values), 0.41 and 25, 0.16 and 40, and 0.43 and 40 for mass fractions and C/N ratios of grain, root and leaf plus stem, respectively, and 3600 °C for TDD. The detailed management practices (Table S1) were obtained from Li et al. (2009) and Liu et al. (2014). Compared with the conventional fertilizer application rate of 110−140 kg N ha−1 yr−1 for the cotton, the fertilizer doses of 2007 and 2008 were reduced to 66−75 kg N ha−1 yr−1 by local farmers to avoid the overgrowth of leaves instead of seeds or lint. The measured data used for validation were available for soil (5 cm depth) temperature, topsoil (0−6 cm) moisture and N2O and NO emissions in 2007−2009 (Liu et al., 2010, 2014), CH4 uptake fluxes during the period from March to November 2010 (unpublished data of the authors), grain yields, and NEE during the period from November 2008 to November 2009 (Wang et al., 2013a). For the W-M, the crop parameters and other inputs used by Cui et al. (2014) were directly adopted in this study." (Please see lines 110-138, lines 150-179 and lines 200-215)

Lines 222-226 what do you want to say? It is not clear to me. Please keep in mind that you are not studying the environmental impacts of using pesticides.

Revised. The statements of pesticides throughout the full manuscript have been deleted so as to keep concentrated as the reviewer suggested.

Discussion Please delete lines 526-527. I do not see that such statement helps to

your work. Unfortunately– your model validation is poor as it evaluates only one site and does not include a rotation of all three commodity crops together. Remember that optimization studies rely on robust site validations. These validation studies should be done using several pilot areas with different geographical, climatic and soil conditions; different types of reference data (long term datasets) used for model calibration. I am not sure that you will get the same results if you apply your best rotation and management practices across different geographical, climatic and soil conditions. A regional simulation will help you to clarify this. I would start with this: The scenario analysis relying on model simulations in this study showed that environmental contamination can be reduced while a) sustaining crop yields and b) increasing carbon sequestration in the soil. Reductions of environmental i.e. N losses are attributed to the better synchronization of crop N requirements and soil N availability......

Revised. The Discussion has been revised as the reviewer suggested so as making it more concise and focused. For instance, "The simulated positive annual changes in SOC for the W-M were mainly attributed to the incorporation of full aboveground residues (at rates of 5.1−7.0 Mg C ha−1 yr−1), which were favorable for carbon sequestration (Han et al., 2016). However, the negative annual changes in SOC for the cotton cropping system resulted from a notable $CO_2$ emission in long fallow season relative to that of the W-M (Liu et al., 2019). As a remarkable carbon sink, the W-M with incorporation of full crop residues could even totally compensate for the SOC lost during the first cotton-planting year following cultivation of the W-M. Thus, the annual change in SOC was generally positive in the first cotton-cultivation year. The rotation patterns of R0 and R1 acted as net GHG sinks since the increased SOC exceeds the increased $N_2O$ emission related to the W-M cultivation, while the others all functioned as net GHG sources. The higher application rate of fertilizer for the W-M than for cotton resulted in more reactive nitrogen remained in the soil (Chen et al., 2014; Ju et al., 2009), thereby stimulating more emissions of the nitrogenous air pollutants and $N_2O$ under the scenarios with fewer cotton planting years. Therefore, the appropriate rotation pattern of cotton and the W-M can allow the sustainable intensification

with maximal yield and economic benefits, balanced soil organic carbon budget and minimal negative impacts on the environment." (Please see lines 363-461)

Lines 531-532 Why do you discuss about pesticides when the DNDC model does not account for?. Please state that DNDC model assumes balanced nutrient supplies for any crops as well as optimum phytosanitary conditions. Hence negative effects of monoculture are not accounted for.

Revised. The statement has been revised based on the reviewer's suggestion. "For the cotton, period of 5 consecutive years is usually applied for the longest cotton monoculture to stabilize its yields. Meanwhile, balanced elemental nutrients have been applied in cotton cultivation, and thus the negative effect of monoculture on cotton yields can be offset in practice (Han, 2010). As DNDC model assumes balanced nutrient supplies for any crops as well as optimum phytosanitary conditions, the negative effects of monoculture are not accounted for. (e.g., Li, 2017)." (Please see lines 368-372)

suggest you to add an uncertainty section as requested before.

Revised. The section on model relative uncertainty of the validation was added, and then we provided the uncertainty of the BMP, which reflect the impacts of validation on BMP. "In this study, the MRB of simulated variables were regarded as the relative uncertainty of the model validation, which were further used for estimating the relative uncertainty of each scenario based on the error transfer formula (Eqs. (S1-4))." (Please see lines 210-212) and section 3.4"The relative uncertainty resulted from the model validation was calculated based on MRB and error transfer formula (Eqs. (S1-4)). The MRB of cumulated $N_2O$, NO, NEE and $CH_4$ were 3%, 6%, 2% and 8%, respectively, which were used for calculating the relative uncertainty of NIP for all 6000 scenarios. For the BMP of each rotation pattern, the scenarios, whose uncertainty ranges had some overlap as that of the BMP, showed no significant differences from each other. Thus, 7, 8, 4, 2 and 2 alternative scenarios were selected for the BMP of R0, R1, R2, R3 and R4, respectively, with the average relative uncertainty of 5.1%. For

the final identified BMP of N90/353_I82_IS_RI90_T10 involved in R3 rotation pattern, the relative uncertainty of the NIP was 4.7%, ranged from 314 to 345 USD ha−1 yr−1. There were another two alternative scenarios (N94/366_I91_IS_RI95_T10 and N97/378_I88_IS_RI70_T5) in R3, which indicated the trade-off effects of different field managements, such as the opposite effect of reduced residue incorporation (decrease âŰşSOC) and tillage depth (increase âŰşSOC) on âŰşSOC. They were also regarded as alternative BMP for the system." (Please see lines 309-321)

Please also note the supplement to this comment:
https://www.biogeosciences-discuss.net/bg-2018-401/bg-2018-401-AC1-supplement.zip

---

## Author Comment (AC2) · 16 Jan 2019

RC2 This manuscript describes a modeling effort to evaluate the biogeochemical effects of optimizing a cotton, wheat, maize rotation and field management practices. This work is within the scope of Biogeosciences. Modeling efforts such as this are important given the difficulty in designing field experiments that adequately capture the appropriate biogeochemical parameters for each treatment. Overall this work is important in increasing our understanding the environmental impacts of management practices. However, one major weakness is a lack of sufficient experimental validation.

Specific comments Page 3, lines 67-68 describe the "release potentials of nitrogenous pollutants". It would be helpful to describe these pollutants in the context of agricultural practices (e.g. nitrate leaching, etc.).

Revised. The sentence has been revised as the reviewer suggested. "High nitrogen and water inputs can result in high release potentials of nitrogenous pollutants and induce a series of environmental problems, such as increased nitrate leaching for water pollution" (Please see lines 55-56)

Page 6, Section 2.2: This description of the DNDC95 model is somewhat redundant with the introduction and could either be shortened here or removed from the introduction.

Revised. The description of DNDC in the Introduction has been deleted and the content in Materials and methods has been shortened. "The DNDC95 model used in this study is one of the latest DNDC versions (www.dndc.sr.unh.edu/model/GuideDNDC95.pdf). The model consists of two components with six modules in total. Driven by given primary ecological factors, the former component simulates the field states of a soil-plant system, such as soil chemical and physical status, vegetation growth and organic matter decomposition. Driven by the soil regulating variables yielded by the former component, the latter component simulates the core biogeochemical processes of carbon and nitrogen transformations and physical processes of liquid and gas transportations and thus the annual dynamics of net ecosystem exchanges of carbon dioxide ($CO_2$) (NEE); emissions of methane ($CH_4$), nitrous oxide ($N_2O$), $NH_3$ and $NO$; and $NO_3-$ leaching and the inter-annual dynamics of SOC and NEGE. These features enable the model to investigate the integrative biogeochemical effects of the changes in rotation patterns of multiple crops and/or other management practices based on comprehensive validation." (Please see lines 89-99)

Page 8, Section 2.4: Six level-I scenarios are described with increasing the number of cotton rotations with W-M. How were multiple cotton crops incorporated? For R2 was it

2 years of cotton followed by 4 years W-M or 1 or 2 years of W-M between each cotton crop?

Revised. The description of level-I scenarios has been improved to avoid ambiguity as the reviewer suggested. "There were six level-I scenarios in total, hereinafter referred to as R0, R1,..., R5, and the number of rotation indicated the years of consecutive cotton planting. For instance, R0 denotes the 6-year monoculture of the W-M; R2 represents the 2-year continuous cotton rotated with the 4-year consecutive W-M; and so on." (Please see lines 142-145)

Page 23, Section 4.3: This discussion of the BMP should also include a discussion of the potential impacts of weed or disease pressure. Continuous cultivation of these crops could lead to challenges for weed or disease management which does not appear to be adequately addressed by the model. This section should include a discussion of these limitations.

Revised. The discussion on weed or disease pressure due to the model limitation has been included as the reviewer suggested. "The first is the possible limitation of the applied model, which cannot simulate the potential effects of monoculture on weed and disease, as well as yields. Continuous cultivation of these crops, especially for cotton, could lead to challenges for weed and disease managements. On the other hand, the scenarios of continuous no-tillage for the W-M could also increase the pressure of weed and disease. But these effects could not be adequately addressed by the current model. Thus, proper parameterization of the effects of monoculture and no-tillage on weed and disease, as well as yields, would be beneficial for screening the more realistic and effective BMP." (Please see lines 430-437)

Page 25: The number of scenarios mentioned in the discussion is different from page 9 and 18.

Revised. Monte Carlo simulation has been applied for BMP selection, thus there are 1000 scenarios for each rotation pattern and 6000 scenarios for BMP selection. "To

screen the BMP of six rotation patterns in interaction with all considered management practices, the variation of fertilizer amount, irrigation amount and residue incorporation rate was set as 40% of baseline to baseline (N44/172 to N110/430), 40% of baseline to baseline (I40 to I100) and 0 to 100% (RI0 to RI100). The factors of irrigated method and tillage consisted of flood (IF) and sprinkle (IS) irrigation, and no-tillage (T0), reduced tillage (5 cm and 10 cm, T5 and T10) and conventional tillage (20 cm, T20), respectively. We assumed the frequency distribution of all the factors were uniform. Monte Carlo simulations, 1000 combination scenarios of field managements, were used to screen the BMP for each rotation pattern, and the final BMP for the system were selected from the BMPs of six rotation patterns in light of 6000 combination scenarios." (Please see lines 161-170)

Figure 2: It is difficult to differentiate between the different rotation patterns since most of the symbols are stacked. An additional figure or table showing the order of rotations for R0-R5 would be helpful.

Revised. An additional table (Table S4) in supplementary material has been provided to differentiate the different rotation patterns as the reviewer suggested. (Please see the Table S4 in supplementary material)

Technical corrections Line 79: Remove "s" from "contents" Line 132: Replace "its" with "associated" Line 619 should not be indented.

Revised. The sentences have been corrected as the reviewer suggested. (Please see line 63, line 86 and line 430)

Please also note the supplement to this comment:
https://www.biogeosciences-discuss.net/bg-2018-401/bg-2018-401-AC2-supplement.zip

---

## Author Response (AR1)

Comments to the Author:
Both Referees provided insightful and detailed reviews and noted that validation in particular could be improved. Please carefully address all comments by reviewers and create a revised manuscript with major revisions and I would be happy to reconsider the manuscript for publication.

**Revised.** We have revised the manuscript one by one as the editor and reviewers suggested, especially for the reminder of improvement for model validation. In addition, an additional co-author (*Jiang Zhu; jzhu@mail.iap.ac.cn*) was added in the revised manuscript.

RC1
The manuscript entitled "Modeling the biogeochemical effects of rotation pattern and field management practices in a multi-crop (cotton, wheat, maize) rotation system: a case study in northern China" is within the scope of BG. To ensure reliability, models should be tested and improved as part of their development and application. The manuscript is important in that context (though it is poor- it lacks for a 6 years validation that includes a rotation of all three commodity crops as well as all management practices studied in question) but the novelty of this manuscript lies with the optimization of different rotation patterns (of three cultivars: cotton, wheat, maize) and management practices which is very complex. Overall, the manuscript lacks of structure and the English language in the manuscript needs to be improved. The manuscripts need major revisions to be acceptable for the publication. In the current state should be rejected.

**Revised.** We have revised the manuscript one by one as the reviewer suggested. The model has been modified and improved for the NO simulation, and further tested and validated with the observations in two adjacent plots of cotton winter wheat-summer maize double-cropping systems. The manuscript has been new structured, especially for the Introduction, and the English language has been revised by AJE. Monte Carlo simulation for BMP identification and model uncertainty for validation were performed and discussed as the reviewer suggested. Please see the detailed revision notes below.

For more details please see my comments below:
In the site simulation NEE and NO emission are predicted with lower accuracy by the model, then how this impacted the optimization of mitigation options?

**Revised.** The model has been modified to improve the daily NO simulation by incorporating the effect of soil moisture on NO production during nitrification. "*To improve the model performance during daily NO simulations for cotton cropping system, the production process for NO ($NO_p$) during nitrification was modified. For the original version by Cui et al. (2014), the NO production was simply quantified as a fixed fraction (0.003) of nitrification (Fni), which reflected a constant*

*production rate of NO during nitrification. For the modified version, the effect of the soil moisture (SM in water-filled pore space (WFPS)) on NO production was incorporated in view of that applied during the $N_2O$ production process in nitrification (Eq. (1)). The maximum NO production rate ($K_n$) was calibrated as 0.03 using the observed daily NO fluxes from 2007−2008 for the cotton cropping system. The incorporated soil moisture effects indicated that high soil moisture facilitated the production of NO during nitrification under the concept of an "anaerobic balloon".*" (Please see lines 111-119). "*In comparison, the modified model significantly improved the model performance for the daily NO fluxes, especially for emissions in the spring. The IA, NSI, and ZIR slope and the $R^2$ values increased from 0.62 to 0.78, -1.03 to -0.04, 0.37 to 0.54 and 0.09 to 0.39, respectively.*" (Please see lines 250-252).

"*According to the BMP screening method used in this study, a solid validation of the cumulative emissions of $N_2O$, NO, NEE and $CH_4$ was the basis for identifying the BMP, rather than those at the daily scale.*" (Please see lines 214-216). "*To screen the BMP of the cotton and W-M rotation system under various management practices based on the annual simulation results, the model performances for the cumulative $N_2O$, NO, NEE and $CH_4$ were required for validation. According to the updated results of Cui et al. (2014) and this study using the modified model, the model showed satisfactory performances for simulating the cumulative variables, with ZIR slope and the $R^2$ values of 0.90 and 0.83 (n = 12, P < 0.001), 0.90 and 0.94 (n = 11, P < 0.001), 0.98 and 0.99 (n = 5, P < 0.001), and 0.99 and 0.91 (n = 7, P < 0.001) for the cumulative $N_2O$, NO, NEE and $CH_4$, respectively, which provided a solid basis for the BMP identification at this site scale (Fig. 2). These results suggested that the DNDC95 model could be applicable in investigating the biogeochemical effects of different rotation patterns between the cotton and W-M and the effects of different management practices.*" (Please see lines 268-277). "*For the cotton in this study, the underestimated daily NO fluxes in the spring in the model from Cui et al. (2014) was improved by the modified model. However, the improvements in the daily NO fluxes did not significantly affect the annual cumulative emissions, which were not major contributors to the annual emissions (Liu et al., 2015).*" (Please see lines 375-379). "*However, this defect did not result in significant biases in the cumulative NEE. Therefore, the identified BMPs that depended on the simulated annual NEE were reliable.*" (Please see lines 383-385).

The relative uncertainty of the model validation was evaluated, and then we provided the uncertainty of the BMP, which reflect the impacts of validation on BMP. "*In addition, the relative uncertainty resulting from the model validation was calculated based on the MRB and error transfer formula (Eqs. (S1-4)). The MRBs of the cumulative $N_2O$, NO, NEE and $CH_4$ for cotton and W-M were 2% and 8%, 11% and 11%, 10% and 4%, and 2% and 2%, respectively, and the MRB of the cumulative $NH_3$ for W-M was 6%. These percentages were used to calculate the relative uncertainty of the NIP for all 6000 scenarios. For the BMP of each rotation pattern, the scenarios, for which the uncertainty ranges had some overlap with that of the BMP, showed no significant differences from one another. Thus, 6, 7, 4, 3 and 0 alternative scenarios were selected for the BMPs of $R_0$, $R_1$, $R_2$, $R_3$ and $R_4$, respectively, with an average relative uncertainty of 3.7%. For the final identified BMP of N90/353_I82_IS_RI90_T10 involved in the $R_3$ rotation pattern, the relative uncertainty of the NIP was 3.1%, ranging from 317 to 338 USD $ha^{-1}$ $yr^{-1}$. There were three other alternative scenarios (N94/366_I94_IS_RI75_T20, N94/366_I91_IS_RI95_T10 and N97/378_I88_IS_RI70_T5) in $R_3$, which indicated the trade-off effects of different field managements, such as the opposite effect of reduced residue incorporation (decrease $\Delta SOC$) and*

*tillage depth (increase ΔSOC) on the ΔSOC. These scenarios were also regarded as alternative BMPs for the system (Table 1).*" (Please see lines 343-357).

The novelty of this manuscript lies with optimization of mitigation options at site level but authors exploited this inadequately in this manuscript. Elaborating and extending optimization analysis will add substantial knowledge and value to the manuscript. What about using i.e. Monte Carlo optimization technique to screen different set of possible agricultural management practices (a multiple optimization criteria that includes crop rotation in interaction with all studied management practices) which maximize yields while minimizing environmental effects.

**Revised.** Monte Carlo technique has been applied for identifying the BMP. "*To screen the BMP of six rotation patterns in the interaction with all the considered management practices, the variation in the fertilizer amount, irrigation amount and residue incorporation rate was set as 40% of the baseline to the baseline (N44/172 to N110/430), 40% of the baseline to the baseline (I40 to I100) and 0 to 100% (RI0 to RI100). The irrigated method and tillage factors consisted of flood (IF) and sprinkle (IS) irrigation and no-tillage (T0) and reduced tillage (5 cm and 10 cm, T5 and T10) for W-M and conventional tillage (20−30 cm, T20). We assumed that the frequency distributions of all the factors were uniform. Monte Carlo simulations, at 1000 combined field management scenarios, were used to screen the BMP for each rotation pattern, and the final BMP was selected from the BMPs of six rotation patterns in light of the 6000 combined scenarios.*" (Please see lines 177-186).

Uncertainty quantification is a critical challenge in both validation and calibration. There is NO mention of model uncertainty in the manuscript. I suggest adding one section on model uncertainty and discussing uncertainties and how that might propagate to model outputs in this study. Authors should also focus on potential applications of optimization considering uncertainty. Otherwise these mitigation options have only academic interest and not much real-world value. Please, see the specific comments below.

**Revised.** The section on model relative uncertainty of the validation was added. The model uncertainty for validation was reduced due to the model modification and then we provided the uncertainty of the BMP, which reflect the impacts of validation on BMP. In addition, three other BMP alternatives were screened based on relative uncertainty. "*In this study, the MRBs of cumulative $N_2O$, NO, NEE and $CH_4$ were regarded as the relative uncertainty of the model validation, which were further used for estimating the relative uncertainty of each scenario based on the error transfer formula (Eqs. (S1−4)).*" (Please see lines 228-230). "*Compared with the simulation results by the model version of Cui et al. (2014), the $R^2$ of ZIR by the modified model for the cumulative $N_2O$, NO, NEE and $CH_4$ increased by 0−8%, and thus reduced the model uncertainty for validation at an annual scale. In addition, the relative uncertainty resulting from the model validation was calculated based on the MRB and error transfer formula (Eqs. (S1-4)). The MRBs of the cumulative $N_2O$, NO, NEE and $CH_4$ for cotton and W-M were 2% and 8%, 11% and 11%, 10% and 4%, and 2% and 2%, respectively, and the MRB of the cumulative $NH_3$ for W-M was 6%. These percentages were used to calculate the relative uncertainty of the NIP for all*

*6000 scenarios. For the BMP of each rotation pattern, the scenarios, for which the uncertainty ranges had some overlap with that of the BMP, showed no significant differences from one another. Thus, 6, 7, 4, 3 and 0 alternative scenarios were selected for the BMPs of $R_0$, $R_1$, $R_2$, $R_3$ and $R_4$, respectively, with an average relative uncertainty of 3.7%. For the final identified BMP of N90/353_I82_IS_RI90_T10 involved in the $R_3$ rotation pattern, the relative uncertainty of the NIP was 3.1%, ranging from 317 to 338 USD $ha^{-1}$ $yr^{-1}$. There were three other alternative scenarios (N94/366_I94_IS_RI75_T20, N94/366_I91_IS_RI95_T10 and N97/378_I88_IS_RI70_T5) in $R_3$, which indicated the trade-off effects of different field managements, such as the opposite effect of reduced residue incorporation (decrease $\Delta SOC$) and tillage depth (increase $\Delta SOC$) on the $\Delta SOC$. These scenarios were also regarded as alternative BMPs for the system (Table 1).*" (Please see lines 341-357).

Introduction: In general I would say that the introduction is too long and not enough focused on the task. There are plenty of paragraphs which must be shortened and better structured. This will improve the content and impact of the current manuscript. Please skip unnecessary things. i.e. frequent applications of pesticides and/or herbicides. My suggestion is to reduce the introduction section to max. 2 pages.

I will start with one example: Globally, fiber crops (i.e. cotton) and cereals such as wheat and maize have been playing a relevant role in humanity as they are a primary source for the textile and food industry. In China, while the cultivation of cotton only covers between 2.0−3.9% of the annual crop harvest areas (cotton lint production of 5.3−7.6 million metric tons during 2007−2016), the cultivation of cereals is significantly large. Wheat and maize account for 39% and 26% of the harvest area and represent 129 and 220 million metric tons of grain in 2016, respectively (China Statistical Yearbook, 2017).

Northern China is not only the second most important area of cotton production but the largest region of the winter wheat−summer maize double-cropping system (i.e., both crops harvested within a year, hereinafter referred to as W-M) in the country (e.g., Cui et al., 2014). Crop rotations of cotton and the W-M have been commonly applied in this region (e.g., Liu et al., 2010, 2014) and are typically alternated every 3−5 years. During the last decades, cotton, wheat and maize's yields have increased by means of intensified agricultural management practices such as: increased fertilizer inputs, advanced irrigation methods (Han, 2010). A recent study (Liu et al., 2019) indicated that the cotton cropping system in northern China persistently functioned as an intensive carbon or net greenhouse gas (GHG) source compared to the W-M because of strong carbon dioxide ($CO_2$) emissions during the long non-growing periods. Add Reference. revealed that the change in storage of soil organic carbon ($\Delta SOC$), net ecosystem GHG emission (NEGE) and other biogeochemical processes of the multiple-cropping systems in northern China likely are closely related to the rotation pattern of cotton and the W-M. Thus, one can hypothesize that identifying and adopting optimal rotation pattern of cotton and the W-M are beneficial for soil carbon sequestration and mitigation of GHG emissions in the region...... Please see general comment of this section!

**Revised.** The Introduction has been structured and revised based on reviewer's comments, and the length has been reduced to less than 2 pages, with words reduction from 1400 to 850. For instance,

*"An objective method is applied to identify the best management practice (BMP), which evaluates each decision variable with price-based proxies or other measures and screens the best option with the minimal negative impact potential (NIP) under the given constraints at an annual scale (e.g., Cui et al., 2014; Xu et al., 2017). To screen the BMP using the multi-goal approach, it is essential to quantify the biogeochemical effects of management practices at an annual scale. As field experiments often focus only on the decision variables of very few management practices during short periods (e.g., Ding et al., 2007; Liu et al., 2010, 2015; Wang et al., 2013a, b), the process-oriented biogeochemical models have the potential to overcome this limitation, through models such as the DeNitrification-DeComposition (DNDC) (e.g., Chen et al., 2016; Giltrap et al., 2010; Li, 1992, 2000; Zhang et al., 2017a), DAYCENT (e.g., Delgrosso et al., 2005) and LandscapeDNDC (e.g., Haas et al., 2012).*" (Please see lines 33-87).

Material and Methods

General comment: Same as above, please shorten and restructure this section Put sections 2.1 and 2.3 together (short and concise)

**Revised.** We shorten and revised the content of Materials and methods as the reviewer suggested. Put the sections 2.1 and 2.3 together as section 2.2 and reconstructed. For instance, "*The modified model was validated in the plot cultivated with cotton. The daily meteorological data from 2004−2010 were obtained directly from Cui et al. (2014). The measured data were used directly for the minimum required soil properties. The input data on the field capacity and wilting point in the WFPS were 0.65 and 0.2, respectively, as cited from Cui et al. (2014). The crop parameters for cotton were directly determined by the field measurements, which were 1900 kg C ha$^{-1}$ for potential grain (1.2 times the mean of the measured values), 0.41 and 25, 0.16 and 40, and 0.43 and 40 for the mass fractions and C/N ratios of the grain, root and leaf plus stem, respectively, and 3600 ℃ for the TDD. Detailed management practices (Table S1) were obtained from Li et al. (2009) and Liu et al. (2014). Compared with the conventional fertilizer application rate of 110−140 kg N ha$^{-1}$ yr$^{-1}$ for the cotton, the fertilizer doses for 2007 and 2008 were reduced to 66−75 kg N ha$^{-1}$ yr$^{-1}$ by local farmers to avoid the overgrowth of the leaves in place of seeds or lint. The measured data used for calibration and validation were available for the soil (5 cm depth) temperature, topsoil (0−6 cm) moisture and N$_2$O and NO (the daily NO fluxes from 2007−2008 were used for the K$_n$ calibration and the data from 2008−2009 were used for validation) emissions from 2007−2009 (Liu et al., 2010, 2014), CH$_4$ uptake fluxes (Liu et al., 2019), grain yields, and NEE (Liu et al., 2019). In addition, because the model has been modified from the version used by Cui et al. (2014) for W-M in the adjacent plot, the W-M simulations were performed with the modified version again, using the crop parameters and other inputs adopted by Cui et al. (2014). The cotton and W-M validation data are detailed in Table S2. Thus, to identify the BMP, both the validations of cotton and W-M for the cumulative N$_2$O, NO and CH$_4$ under various field managements, NH$_3$ and NEE were also analyzed in this study.*" (Please see lines 89-195).

Lines 222-226 what do you want to say? It is not clear to me. Please keep in mind that you are not studying the environmental impacts of using pesticides.

**Revised.** The statements of pesticides throughout the full manuscript have been deleted so as to keep concentrated as the reviewer suggested.

Discussion

Please delete lines 526-527. I do not see that such statement helps to your work. Unfortunately– your model validation is poor as it evaluates only one site and does not include a rotation of all three commodity crops together. Remember that optimization studies rely on robust site validations. These validation studies should be done using several pilot areas with different geographical, climatic and soil conditions; different types of reference data (long term datasets) used for model calibration. I am not sure that you will get the same results if you apply your best rotation and management practices across different geographical, climatic and soil conditions. A regional simulation will help you to clarify this.

I would start with this: The scenario analysis relying on model simulations in this study showed that environmental contamination can be reduced while a) sustaining crop yields and b) increasing carbon sequestration in the soil. Reductions of environmental i.e. N losses are attributed to the better synchronization of crop N requirements and soil N availability......

**Revised.** The Discussion has been revised as the reviewer suggested so as making it more concise and focused. For instance, "*The scenario analysis relying on model simulations in this study showed that environmental contamination can be reduced while i) sustaining crop yields, ii) increasing soil carbon sequestration and iii) decreasing the net ecosystem GHG emissions. Reductions in environmental contamination are attributed to the better synchronization of crop nitrogen requirements and soil nitrogen availability. For cotton, a period of 5 consecutive years is usually applied as the longest cotton monoculture to stabilize its yields. In addition, balanced elemental nutrients have been applied during cotton cultivation, and thus the negative effect of monoculture on cotton yields can be offset in practice (Han, 2010). Because the DNDC model assumes balanced nutrient supplies for any crops as well as optimum phytosanitary conditions, the negative effects of monoculture are not accounted for here (e.g., Li, 2017). The simulated positive annual changes in the SOC for W-M were mainly attributed to the incorporation of the full aboveground residues (at rates of 5.1−7.0 Mg C ha$^{-1}$ yr$^{-1}$), which favored for carbon sequestration (Han et al., 2016). However, the negative annual changes in the SOC for the cotton cropping system resulted from notable CO$_2$ emissions over a long fallow season relative to that of W-M (Liu et al., 2019). As a remarkable carbon sink, the W-M under the incorporation of the full crop residues could completely compensate for the SOC lost during the first cotton-planting year following the W-M cultivation. Thus, the annual change in the SOC was generally positive during the first cotton cultivation year. The rotation patterns of R$_0$ acted as net GHG sinks since the increased SOC exceeds the increased N$_2$O emission related to W-M cultivation, while the others all functioned as net GHG sources. The higher fertilizer application rate for W-M than for cotton resulted in the more reactive nitrogen remaining in the soil (Chen et al., 2014; Ju et al., 2009), thereby stimulating higher emissions of nitrogenous air pollutants and N$_2$O in the trials with fewer cotton cultivation years. Therefore, the appropriate rotation pattern of cotton and W-M can realize sustainable intensification with maximum yield and economic benefits, a balanced soil organic*

*carbon budget and minimal negative impacts on the environment."* (Please see lines 359-494). The confirmation of BMP in this region required further validation. *"In addition, the alternative scenarios with overlapping ranges of uncertainty were also chosen as BMP alternatives, although the increase in fertilizer and irrigation increased the negative effects to some extent. Because the identified BMP was based on the sufficient validation only at this site, it should be the potential BMP in this region, but additional validations for other sites in this region are still required to confirm the BMP."* (Please see lines 458-463).

Lines 531-532 Why do you discuss about pesticides when the DNDC model does not account for?. Please state that DNDC model assumes balanced nutrient supplies for any crops as well as optimum phytosanitary conditions. Hence negative effects of monoculture are not accounted for.

**Revised.** The statement has been revised based on the reviewer's suggestion. *"For cotton, a period of 5 consecutive years is usually applied as the longest cotton monoculture to stabilize its yields. In addition, balanced elemental nutrients have been applied during cotton cultivation, and thus the negative effect of monoculture on cotton yields can be offset in practice (Han, 2010). Because the DNDC model assumes balanced nutrient supplies for any crops as well as optimum phytosanitary conditions, the negative effects of monoculture are not accounted for here (e.g., Li, 2017)."* (Please see lines 400-404).

suggest you to add an uncertainty section as requested before.

**Revised.** The section on model relative uncertainty of the validation was added. The model uncertainty for validation was reduced due to the model modification and then we provided the uncertainty of the BMP, which reflect the impacts of validation on BMP. In addition, three other BMP alternatives were screened based on relative uncertainty. *"In this study, the MRBs of cumulative $N_2O$, NO, NEE and $CH_4$ were regarded as the relative uncertainty of the model validation, which were further used for estimating the relative uncertainty of each scenario based on the error transfer formula (Eqs. (S1−4))."* (Please see lines 228-230). *"Compared with the simulation results by the model version of Cui et al. (2014), the $R^2$ of ZIR by the modified model for the cumulative $N_2O$, NO, NEE and $CH_4$ increased by 0−8%, and thus reduced the model uncertainty for validation at an annual scale. In addition, the relative uncertainty resulting from the model validation was calculated based on the MRB and error transfer formula (Eqs. (S1-4)). The MRBs of the cumulative $N_2O$, NO, NEE and $CH_4$ for cotton and W-M were 2% and 8%, 11% and 11%, 10% and 4%, and 2% and 2%, respectively, and the MRB of the cumulative $NH_3$ for W-M was 6%. These percentages were used to calculate the relative uncertainty of the NIP for all 6000 scenarios. For the BMP of each rotation pattern, the scenarios, for which the uncertainty ranges had some overlap with that of the BMP, showed no significant differences from one another. Thus, 6, 7, 4, 3 and 0 alternative scenarios were selected for the BMPs of $R_0$, $R_1$, $R_2$, $R_3$ and $R_4$, respectively, with an average relative uncertainty of 3.7%. For the final identified BMP of N90/353_I82_IS_RI90_T10 involved in the $R_3$ rotation pattern, the relative uncertainty of the NIP was 3.1%, ranging from 317 to 338 USD $ha^{-1}$ $yr^{-1}$. There were three other alternative scenarios*

*(N94/366_I94_IS_RI75_T20, N94/366_I91_IS_RI95_T10 and N97/378_I88_IS_RI70_T5) in $R_3$, which indicated the trade-off effects of different field managements, such as the opposite effect of reduced residue incorporation (decrease $\Delta SOC$) and tillage depth (increase $\Delta SOC$) on the $\Delta SOC$. These scenarios were also regarded as alternative BMPs for the system (Table 1).*" (Please see lines 341-357).

RC2

This manuscript describes a modeling effort to evaluate the biogeochemical effects of optimizing a cotton, wheat, maize rotation and field management practices. This work is within the scope of Biogeosciences. Modeling efforts such as this are important given the difficulty in designing field experiments that adequately capture the appropriate biogeochemical parameters for each treatment. Overall this work is important in increasing our understanding the environmental impacts of management practices. However, one major weakness is a lack of sufficient experimental validation.

Specific comments Page 3, lines 67-68 describe the "release potentials of nitrogenous pollutants". It would be helpful to describe these pollutants in the context of agricultural practices (e.g. nitrate leaching, etc.).

**Revised.** The sentence has been revised as the reviewer suggested. "*High nitrogen and water inputs can result in high release potentials for nitrogenous pollutants, and they can induce a series of environmental problems, such as increased nitrate ($NO_3^-$) leaching for water pollution (e.g., Collins et al., 2016).*" (Please see lines 55-57).

Page 6, Section 2.2: This description of the DNDC95 model is somewhat redundant with the introduction and could either be shortened here or removed from the introduction.

**Revised.** The description of DNDC in the Introduction has been deleted and the content in Materials and methods has been shortened. "*The DNDC95 model used in this study is one of the latest DNDC versions (www.dndc.sr.unh.edu/model/GuideDNDC95.pdf). This model consists of two components with six modules in total. Driven by the given primary ecological factors, the former component simulates the field states of a soil-plant system, such as the soil chemical and physical status, vegetation growth and organic matter decomposition. Driven by the soil-regulating variables yielded by the former component, the latter component simulates the core biogeochemical processes of carbon and nitrogen transformations and the physical processes of liquid and gas transportations and thus the annual dynamics of net ecosystem exchanges of carbon dioxide ($CO_2$) (NEE); emissions of methane ($CH_4$), nitrous oxide ($N_2O$), $NH_3$ and NO; and $NO_3^-$ leaching and the inter-annual dynamics of SOC and NEGE. These features enable the model to investigate the integrative biogeochemical effects of the rotation patterns of multiple crops and/or other management practices based on comprehensive validation.*" (Please see lines

90-101).

Page 8, Section 2.4: Six level-I scenarios are described with increasing the number of cotton rotations with W-M. How were multiple cotton crops incorporated? For R2 was it 2 years of cotton followed by 4 years W-M or 1 or 2 years of W-M between each cotton crop?

**Revised.** The description of level-I scenarios has been improved to avoid ambiguity as the reviewer suggested. "*There were six level-I scenarios in total, which are hereinafter referred to as $R_0$, $R_1$..., $R_5$, and the number of the rotation indicated the years of consecutive cotton planting. For instance, $R_0$ denotes a 6-year monoculture of W-M; $R_2$ represents 2-year continuous cotton cropping rotated with 4 years of consecutive W-M; etc.*" (Please see lines 156-160).

Page 23, Section 4.3: This discussion of the BMP should also include a discussion of the potential impacts of weed or disease pressure. Continuous cultivation of these crops could lead to challenges for weed or disease management which does not appear to be adequately addressed by the model. This section should include a discussion of these limitations.

**Revised.** The discussion on weed or disease pressure due to the model limitation has been included as the reviewer suggested. "*The first is the possible limitation of the applied model, which cannot simulate the potential effects of monoculture on weeds and diseases, as well as yields. The continuous cultivation of these crops, especially for cotton, could lead to challenges in weed and disease management. In addition, the continuous no-tillage scenarios for W-M could also increase the weed and disease pressure. However, these effects could not be adequately addressed by the current model. Thus, the proper parameterization of the monoculture and no-tillage effects on the weeds and diseases as well as the yields would be beneficial for screening the more realistic and effective BMP.*" (Please see lines 465-472).

Page 25: The number of scenarios mentioned in the discussion is different from page 9 and 18.

**Revised.** Monte Carlo simulation has been applied for BMP selection, thus there are 1000 scenarios for each rotation pattern and 6000 scenarios for BMP selection. "*To screen the BMP of six rotation patterns in the interaction with all the considered management practices, the variation in the fertilizer amount, irrigation amount and residue incorporation rate was set as 40% of the baseline to the baseline (N44/172 to N110/430), 40% of the baseline to the baseline (I40 to I100) and 0 to 100% (RI0 to RI100). The irrigated method and tillage factors consisted of flood (IF) and sprinkle (IS) irrigation and no-tillage (T0) and reduced tillage (5 cm and 10 cm, T5 and T10) for W-M and conventional tillage (20−30 cm, T20). We assumed that the frequency distributions of all the factors were uniform. Monte Carlo simulations, at 1000 combined field management scenarios, were used to screen the BMP for each rotation pattern, and the final BMP was selected from the BMPs of six rotation patterns in light of the 6000 combined scenarios.*" (Please see lines 177-186).

Figure 2: It is difficult to differentiate between the different rotation patterns since most of the symbols are stacked. An additional figure or table showing the order of rotations for R0-R5 would be helpful.

**Revised.** An additional table (Table S4) in supplementary material has been provided to differentiate the different rotation patterns as the reviewer suggested. (Please see the Table S4 in supplementary material).

Technical corrections Line 79: Remove "s" from "contents" Line 132: Replace "its" with "associated" Line 619 should not be indented.

**Revised.** The sentences have been corrected as the reviewer suggested. (Please see line 63, line 87 and line 465).

---

## Author Response (AR2)

Dear Editor,

As stated in the previous review, the manuscript entitled "Modeling the biogeochemical effects of rotation pattern and field management practices in a multi-crop (cotton, wheat, maize) rotation system: a case study in northern China" is within the scope of BG. To ensure reliability, models should be tested and improved as part of their development and application. The manuscript is important in that context but the important part of this manuscript lies with the optimization of different rotation patterns (of three cultivars: cotton, wheat, maize) and management practices which is very complex. The manuscript has been revised and improved. Nonetheless, there are important gaps (see below) which should be seriously considered. The current manuscript is not acceptable for publication in BG as it is poor in terms of validation as well as in novelty. For more details please see my comments below:

GAPS:
-    it lacks for a 6 years validation that includes a rotation of all three commodity crops as well as all studied management practices in question

**Yes, ideally the model should be validated with a 6-year rotation of all three commodity crops as well as all studied management practices in question. It is unfortunately that such a dataset is still lacking, even though the dataset involved in this study has been the most complete one so far available for the three-crop system. Due to the shortage of financial resources for the very expensive field observations of the trace gases emissions, the experimental period was limited to only three years, during which all the constraint and decision variables were observed for almost all the management practices in question at least in the wheat-maize fields (in practice, some observations failed to obtain expected data due to some technical failures). Three-year observations for the currently applied management practices were simultaneously conducted consecutively in the two neighbor lands cultivated with cotton and wheat-maize, respectively (see Table S3). Because the experimental cotton was in its 3$^{rd}$ to 5$^{th}$ consecutive year of monoculture following the transition from the previous wheat-maize cultivation while the experimental wheat-maize was also in its 3$^{rd}$ to 5$^{th}$ consecutive year following the transition from the previous cotton cultivation, both were assumed to be representative for the cotton or wheat-maize within a 6-year rotation cycle when their observations were used to validate the model. Nevertheless, we still emphasizes at the end of the paper that in the future study it is necessary for a 6-year model validation that includes a rotation of all three commodity crops as well as all studied management practices in question (see lines 690−692 in the revision).**

-    In the site simulation, NEE is still predicted with lower accuracy by the model; then how this impacted the optimization of mitigation options? Furthermore, decision variables such as NH3 volatilization is not validated by observations.

**As a response to this question, in this revision the algorithm of NEE in DNDC was modified (see lines 147−166 and Eqs. 2−6 in subsection 2.2). The model modification significantly improved the simulation accuracy of daily NEE (see lines 346−351 in subsection 3.1 and Figures 1c−e in the revision). Regarding the cumulative NEE for two years of the cotton field, and three crop seasons of the wheat-maize field, the modified model showed very good performance compared to the original model (see lines 378−383 in subsection 3.2 and Figure**

**2b in the revision). For the three full-year cumulative NEE (two for the cotton and one for the wheat-maize field), the modified model simulations showed model relative biases (MRBs) of −13% to 8%, which was much smaller than the reported uncertainty (25% as one times standard deviation) of the eddy covariance observations. Relying on the five annual/seasonal cumulative NEE, which were derived from the eddy covariance measurements in the fields with the crop residues fully retained, and the corresponding observed crop yields, the five annual/seasonal changes in soil organic carbon stock (△SOC) were estimated following the ecosystem carbon balance approach (or according to the mass conservation law), which showed statistically significant agreement with the modified model simulations (see lines 392−398 in subsection 3.2 and Figure 2d in the revision). Further, the corresponding five annual/seasonal net ecosystem aggregate greenhouse gas emissions (NEGEs) were also estimated based on the observation-derived estimates of △SOC and measured methane and nitrous oxide fluxes, which showed significantly consistence with the modified model simulations (see lines 384−391 in subsection 3.2 and Figure 2c in the revision). Regarding the NH$_3$ volatilizations, the model validation in this revision further involved two urea top-dressing events in addition to the previously used one with reported cumulative volatilization during 11 days following the fertilizer amendments. Using the observed cumulative NH$_3$ volatilization of the three urea application events in the wheat-maize fields, the validation showed small MRBs of −9% to 4%, which were much smaller than the double CVs of the spatially replicated measurements (see lines 419−426 in subsection 3.2 and Figure 2h in the revision). Relying on these model errors resulted from the validations using sub-year measurement-derived estimates or event-based observation with a marginally small sample size to screen the BMP alternatives would inevitably influence the accuracy of the results. We are aware of such an insufficiency of this study duo to the insufficient dataset available for the model validation. Hence, in the last section of the paper we emphasize the necessity of comprehensive observations in the future studies to cover all the constraint and decision variables and other factors as well as the crops and management practices in question (see lines 690−692 in the revision).**

- There are some vias between modeled annual no3 leaching and assumed observations for no3 leaching. You stated "For the simulations of other nitrogen losses from the cotton field, the NO3−leaching accounted for 9−12% of the applied fertilizer nitrogen for model validation, which was comparable with the field measurements of 16−17%". This is around 10 Kg N year, rigth? Please, think about what that would means if you scale this value on a regional level (for the whole Northern China) up.

**The validation of the modified model still showed large MRBs of −32% to −27%, which were less than the two times CVs of the spatially replicated field observations for the annually cumulative nitrate leaching. These MRBs represented the model-underestimations by respectively 3−4 and 13−21 kg N ha$^{-1}$ yr$^{-1}$ for the annual nitrate leaching rates in the cotton and wheat-maize fields subject to the currently applied field management practices (see lines 427−433 in subsection 3.2). Understandably, it would be problematic if these underestimations were directly up-scale to the entire northern China region. In fact, these simulation errors in nitrate leaching were also found to overwhelmingly dominant the simulation errors of the NIPs (see lines 626−631 in subsection 4.3 and Table 1 in the revision). However, in this study we did not attempt to make model modification to reduce these MRBs**

**in the nitrate leaching simulations. This is because we were hard to judge whether there were insufficiencies in the scientific structures or inappropriate parameters in the model to dominate these large MRBs due to the too large measurement errors (with two times CVs of 109−115%) for the observed annual cumulative quantities of the nitrate leaching with a too small sample size ($n = 2$). In the revision, we added discussion on this problem** **(see lines 631−637 in subsection 4.3)****.**

How all this gaps affected your BMP and NIP calculations?

In order to improve the quality of this study (which is poor in terms of validation and novelty), I suggest you to do a Parameter-Induced Uncertainty Quantification for NEE + NH3 + NO + N2O and NO3 Leaching. The Bayesian framework using a Markov Chain Monte Carlo (MCMC) method, will help you to estimate the joint model parameter distribution and you can obtain and pick up the best parameter set combination (using a cost function based on model fitting parameters) that represent your measurements. After this, you can use Monte Carlo in order to derive the best management practice for the studied site.

**Adding the MRB-based quantification of uncertainties for concerned variables** **(see lines 292−322 in subsection 2.6 and 613−626 in subsection 4.3)****, as our answers to the question "***how all this gaps affected your BMP and NIP calculations***", we used these uncertainties due to the simulation gaps to quantify the NIP uncertainties and thus to define the precision of BMP screening** **(see Table 1 and Figure 4 in the revision)****.**

**Using MCMC to pick up the best parameter set combination would be necessary if there are significant model biases for model outputs of interest relative to observations with sufficient precision while the key internal model parameters dominating the significant biases, as well as their priori distributions, are known. In this study, the key internal parameters dominating the model biases and their priori distributions were unknown and the observational precision were still low for the annual quantities of $\Delta$SOC, NEGE, NH$_3$ volatilization and nitrate leaching due to too small sample sizes ($n =2$ or 3). This situation did not facilitate the MCMC based Bayesian method to do the internal parameter-induced uncertainty quantification for these variables. Fortunately, the validations showed statistically meaningful consistence between the simulations and observations of the constraint variables (crop yield, $\Delta$SOC, NEGE) and the decision variables (NEGE, NH$_3$ volatilization, NO emission and nitrate leaching) used in screening the best management scenarios. In addition, most of the model-input parameters were obtained from field observations at the field site while minor input parameters on the crops were calibrated using observed yields at harvest. In these regards, the MCMC-based Bayesian method was not necessary to be used in this study. Instead, as a response to this reviewer comment, we used the model biases resulted from the validation of the individual variables to quantify the model error of NIP for each scenario and thus determine the BMP screening precision** **(see subsections 2.6 and 3.2, Table 1 and Figure 4 in the revision)****. In addition, we used the Monte Carlo test to quantify the uncertainties induced by the model input uncertainties of soil parameters (four key soil properties) for the concerned variables and the NIP of the individual management scenarios** **(see subsections 2.6, Table 1 and Figure 4 in the revision)****, which were involved in the discussion for the influences exerted by the model simulation errors for decision variables on the simulation errors of the NIPs** **(see lines 613−637 in subsection 4.3)****.**

I still think that a regional inventory for Northern china should be included in the scope of your study (especially because of the importance of the crop rotation cotton, W-M in this region). For this you can pick up the best 50-100 parameters set combinations (derived with the Bayesian framework) together with the BMP (derived with Monte Carlo) and do a Parameter-Induced Uncertainty Quantification of Regional NEE + NH3 + NO + N2O and NO3 Leaching. This can help to policy decision makers to support farmers in Northern China.

**As the reviewer suggested, to expand this case study to a regional inventory is very important for policy decision makers to support farmers in northern China. In fact, this is in our plan for the future study. To fulfill this very important task, however, there are still two big challenges very tough to be solved. One is the lacking of survey data on historically/currently applied field management practices at a spatial resolution higher than the sub-county or even county level. The other is the lacking of comprehensive observations covering all the constraint and decision variables obtained at spatially replicated field sites in northern China. If these two problems are still there, we do not think reliable BMPs could be resolved using the MCMC-based Bayesian approach. In the future, we will devote to the solution of these challenges to some extent while further improving the biogeochemical model, thus gradually approaching to the great goal.**